# Thermal fluctuations of immature SOD1 lead to separate folding and misfolding pathways

Ashok Sekhar[1,2,7]*, Jessica AO Rumfeldt[3], Helen R Broom[3], Colleen M Doyle[3], Guillaume Bouvignies[1,2,4,5,6,7], Elizabeth M Meiering[3]*, Lewis E Kay[1,2,7,8]*

[1]Department of Molecular Genetics, University of Toronto, Toronto, Canada; [2]Department of Biochemistry, University of Toronto, Toronto, Canada; [3]Department of Chemistry, University of Waterloo, Waterloo, Canada; [4]Université Grenoble Alpes, Grenoble, France; [5]Institut de Biologie Structurale, Centre National de la Recherche Scientifique, Grenoble, France; [6]Institut de Biologie Structurale, Commissariat à l'énergie atomique, Grenoble, France; [7]Department of Chemistry, University of Toronto, Toronto, Canada; [8]Program in Molecular Structure and Function, Hospital for Sick Children, Toronto, Canada

**Abstract** Amyotrophic lateral sclerosis (ALS) is a progressive neurodegenerative disease involving cytotoxic conformations of Cu, Zn superoxide dismutase (SOD1). A major challenge in understanding ALS disease pathology has been the identification and atomic-level characterization of these conformers. Here, we use a combination of NMR methods to detect four distinct sparsely populated and transiently formed thermally accessible conformers in equilibrium with the native state of immature SOD1 (apoSOD1$^{2SH}$). Structural models of two of these establish that they possess features present in the mature dimeric protein. In contrast, the other two are non-native oligomers in which the native dimer interface and the electrostatic loop mediate the formation of aberrant intermolecular interactions. Our results show that apoSOD1$^{2SH}$ has a rugged free energy landscape that codes for distinct kinetic pathways leading to either maturation or non-native association and provide a starting point for a detailed atomic-level understanding of the mechanisms of SOD1 oligomerization.

*For correspondence: ashok.sekhar@utoronto.ca (AS); meiering@uwaterloo.ca (EMM); kay@pound.med.utoronto.ca (LEK)

**Competing interests:** The authors declare that no competing interests exist.

## Introduction

Cu, Zn superoxide dismutase (SOD1) is an extensively studied metalloenzyme that has become a paradigm for understanding protein structure and function as well as folding and misfolding associated with disease (*Valentine et al., 2005*). The mature enzymatically active form of this antioxidant protein (Cu$_2$Zn$_2$SOD1$^{S–S}$) is a homodimer comprised of 153-residue monomer subunits, each of which binds 1 Cu and 1 Zn ion and contains one intra-subunit disulfide bond (*Valentine et al., 2005*). Mutations in the *sod1* gene encoding for this enzyme account for ~20% of all familial ALS (fALS) cases, or about 2% of all ALS occurrences (*Robberecht and Philips, 2013*). Despite the link between SOD1 and fALS and the identification of over 150 disease causing mutants, the molecular events leading to disease remain poorly understood. Inclusions containing SOD1 have been observed in patients afflicted with both familial and sporadic ALS, suggesting that both forms of the disease may share a common pathway (*Matsumoto et al., 1995*; *Rotunno and Bosco, 2013*). However, whether cytotoxicity arises from the aggregates or from soluble misfolded conformers further upstream remains unclear. In its metal-bound oxidized form SOD1 is exceptionally stable, with a melting temperature over 90°C (*Stathopulos et al., 2006*) and is highly resistant to aggregation

**eLife digest** Amyotrophic lateral sclerosis (or ALS) is a disease that affects the nerve cells in the brain and spinal cord, such that more and more of these cells die as the disease progresses. Since nerve cells direct muscle movement, people with ALS increasingly lose control of their muscles, and may therefore lose the ability to speak, eat, move, and breathe.

Around one in five people who have an inherited form of ALS also have mutations in a gene that encodes an enzyme called SOD1. This antioxidant enzyme detoxifies harmful chemicals that are the byproducts of normal cellular activity. The mature active form of this enzyme contains two SOD1 proteins, each of which binds one copper ion and one zinc ion. Each protein in a pair also contains a strong bond that helps to stabilize its three-dimensional structure. However, immature copies of this protein, which lack the strong bond and metal ions, often fold into the wrong shape. These misfolded proteins can clump together into clusters, and potentially lead to the development of ALS.

Efforts to study misfolded proteins are limited by the fact that misfolding is a rare event. As a result, protein samples generally contain very small fractions of misfolded molecules and most techniques for investigating protein structure are ill-suited to probe the process of misfolding. Nuclear magnetic resonance spectroscopy (or NMR for short) is one technique that has been used to visualize flexible proteins, and protein folding and misfolding, down to the level of individual atoms. Recent improvements in NMR spectroscopy have opened up the possibility of studying protein structures that exist at levels as low as 1% of the sample.

Now, Sekhar et al. have used NMR to document the dynamics of immature SOD1 proteins. The experiments show that the protein is flexible and readily switches back and forth between its 'default' shape and one of at least four different shapes; and at any one time, the fraction of molecules in solution that were in each of these shapes was around 1%. Two of the shapes have many of the same features as those in a mature SOD1 protein pair. But the other two shapes involve SOD1 proteins interacting in unusual ways, and may resemble the cluster-forming misfolded SOD1 proteins.

By using the NMR data to create three-dimensional models of the SOD1 proteins, Sekhar et al. were able to identify two sites on immature SOD1 proteins involved in these unusual interactions that are inaccessible in mature enzymatically active SOD1. These findings suggest that targeting these two sites with drugs could possibly block the formation of toxic versions of SOD1 early in the course of ALS, and thus prevent the progression of the disease.

($Stathopulos\ et\ al.,\ 2003$; $Sheng\ et\ al.,\ 2012$; $Broom\ et\ al.,\ 2014$). In contrast, the most immature form of SOD1, apoSOD1$^{2SH}$, that lacks metal ions and the disulfide bond has a much higher propensity to misfold and aggregate in vitro (*Furukawa and O'Halloran, 2005*; *Sheng et al., 2012*; *Broom et al., 2014*) and has been hypothesized to be the primary cause of toxicity in vivo (*Zetterström et al., 2013*). Studies of apoSOD1$^{2SH}$ form the basis of the work described herein.

Misfolded protein conformations can be notoriously difficult to characterize because the tools of structural biology are most suited to proteins which adopt only single conformations that are extensively populated. In contrast, misfolded conformers can be high-energy states that are only marginally populated and transiently sampled by the native protein under physiological conditions (*Chiti and Dobson, 2008*). In this case they cannot be observed directly using 'traditional' biophysical approaches. Recent methodological advances in NMR spectroscopy have rendered such previously 'invisible' states (also referred to as excited states in what follows) amenable to structural analysis so long as they exchange with a 'visible' (ground) state, are populated at a level of approximately 0.5% or higher and have lifetimes ranging from 0.5–20 ms (*Palmer et al., 2000*; *Sekhar and Kay, 2013*).

Here, we have used a combination of different spin relaxation NMR experiments to explore the energy landscape of apoSOD1$^{2SH}$ and to elucidate the structural features of several invisible states that are populated via conformational fluctuations originating from the monomeric ground conformer of the protein. Analysis of the resulting data yields structural information about each of the excited states in the form of chemical shifts as well as the kinetics and thermodynamics of the exchange process (*Palmer et al., 2000*; *Korzhnev et al., 2010*; *Nikolova et al., 2011*; *Neudecker et al., 2012*; *Sekhar and Kay, 2013*). We detect four excited state conformers in equilibrium with native

apoSOD1[2SH]. Two of these, involving either the local folding of a helix or dimerization, lead to structural features that are found in the native state of the mature metallated protein. In contrast, the other two conformers correspond to non-native oligomers. Structural models generated from the NMR data highlight the native dimer interface and the electrostatic loop as key motifs in forming non-native intermolecular contacts.

## Results

### ApoSOD1[2SH] is a monomeric β-barrel with dynamic loops

Detailed structural studies have established that each monomer of homodimeric $Cu_2Zn_2SOD1^{S-S}$ forms an 8-stranded antiparallel Greek key β-barrel with two short helices, one each in the Zn-binding (49–80) and electrostatic loops (126–143) (*Figure 1A*) (*Valentine et al., 2005*). As the name implies the Zn-binding loop contains the Zn binding site, one of the cysteines involved in intra-subunit disulfide bond formation, and a large number of residues that make contacts at the native dimer interface. Metal binding is additionally stabilized by residues in the electrostatic loop (*Strange et al., 2003*). Zinc binding, disulfide bond formation, and dimerization limit the flexibility of the Zn-loop and markedly increase protein stability (*Furukawa and O'Halloran, 2005*). A large number of hydrogen bonding interactions in both loops are thought to stabilize the protein by providing a network of connections between the loops and the metal binding sites (*Parge et al., 1992*).

Wild-type (WT) SOD1 contains four Cys residues, C57 and C146 that form the stabilizing intra-subunit disulfide bond described above and non-conserved C6 and C111 that have been replaced by Ala and Ser, respectively (*Lepock et al., 1990*; *Broom et al., 2014*), in the studies we have performed. The $^{1}H$-$^{15}N$ correlation spectrum of C6A,C111S apoSOD1[2SH] (referred to as pWT apoSOD1[2SH] in what follows) is well resolved and characteristic of a folded protein (*Figure 1B*). Protein backbone $^{1}H$, $^{15}N$, and $^{13}C$ chemical shifts provide both structural (*Cavalli et al., 2007*; *Shen et al., 2008*) and dynamical information (*Berjanskii and Wishart, 2005*), with the amplitude of dynamics at each amide site most often quantified through an order parameter squared, $S^2$ (*Lipari and Szabo, 1982*), that can be calculated from the measured chemical shifts. Values of $S^2$ range between 0–1 with the extreme values corresponding to cases of no motion (1) or completely isotropic dynamics (0), respectively. A comparison of the residue-specific order parameters calculated from chemical shifts of the ground states of pWT apoSOD1[2SH] and $Cu_2Zn_2SOD1^{S-S}$ (chemical shifts for the latter form obtained from *Banci et al. (2002a)*) shows that both loops become disordered in apoSOD1[2SH] (*Figure 2A*), while residues involved in β strands remain rigid. An excellent correlation is obtained between predicted inter-strand NOE patterns (*Figure 2B*) based on the X-ray structure of $Cu_2Zn_2SOD1^{S-S}$ (*Strange et al., 2003*) and observed NOEs measured from spectra recorded on apoSOD1[2SH], establishing that the β-barrel remains intact in the immature form of the protein as well. This is further confirmed by measured values of amide proton temperature coefficients in regions where β strands are known to form in $Cu_2Zn_2SOD1^{S-S}$. Values greater than −4.6 ppb/K (red line) denote amides that are hydrogen bonded in structure (*Cierpicki and Otlewski, 2001*), as observed for the great majority of the β strand protons in apoSOD1[2SH] (*Figure 2C*).

We have also established that under the conditions of our NMR experiments pWT apoSOD1[2SH] is monomeric, as expected based on previous characterizations of this protein (*Arnesano et al., 2004*). *Figure 2D* correlates $^{15}N$ transverse relaxation rates measured at protein concentrations of 2 mM and 3.5 mM (red line) or 0.8 mM and 3.5 mM (green), with each point corresponding to a separate amide position in the protein. Slopes of 0.87 and 0.72 are in excellent agreement with expected values if the differences in protein concentration dependent relaxation are due to viscosity, since very similar ratios of NMR derived translational diffusion coefficients are obtained from measurements at different protein concentrations (0.87 and 0.76). Note that concentration dependent oligomerization would lead to deviations between relaxation and diffusion ratios, since these measures scale differently with molecular size ($MW^{-1}$ and $MW^{-1/3}$ for relaxation times and diffusion, respectively, where MW is the molecular weight of the assumed spherical particle [*Cantor and Schimmel, 1980*]). Notably, isotropic rotational correlation times were measured to be close to a factor of two different between pWT apoSOD1[2SH] and $Cu_2Zn_2SOD1^{S-S}$ (11.6 ns and 18.2 ns, 'Materials and methods'), although an exact twofold difference would not be expected because of the distinct shapes of monomeric and dimeric SOD1.

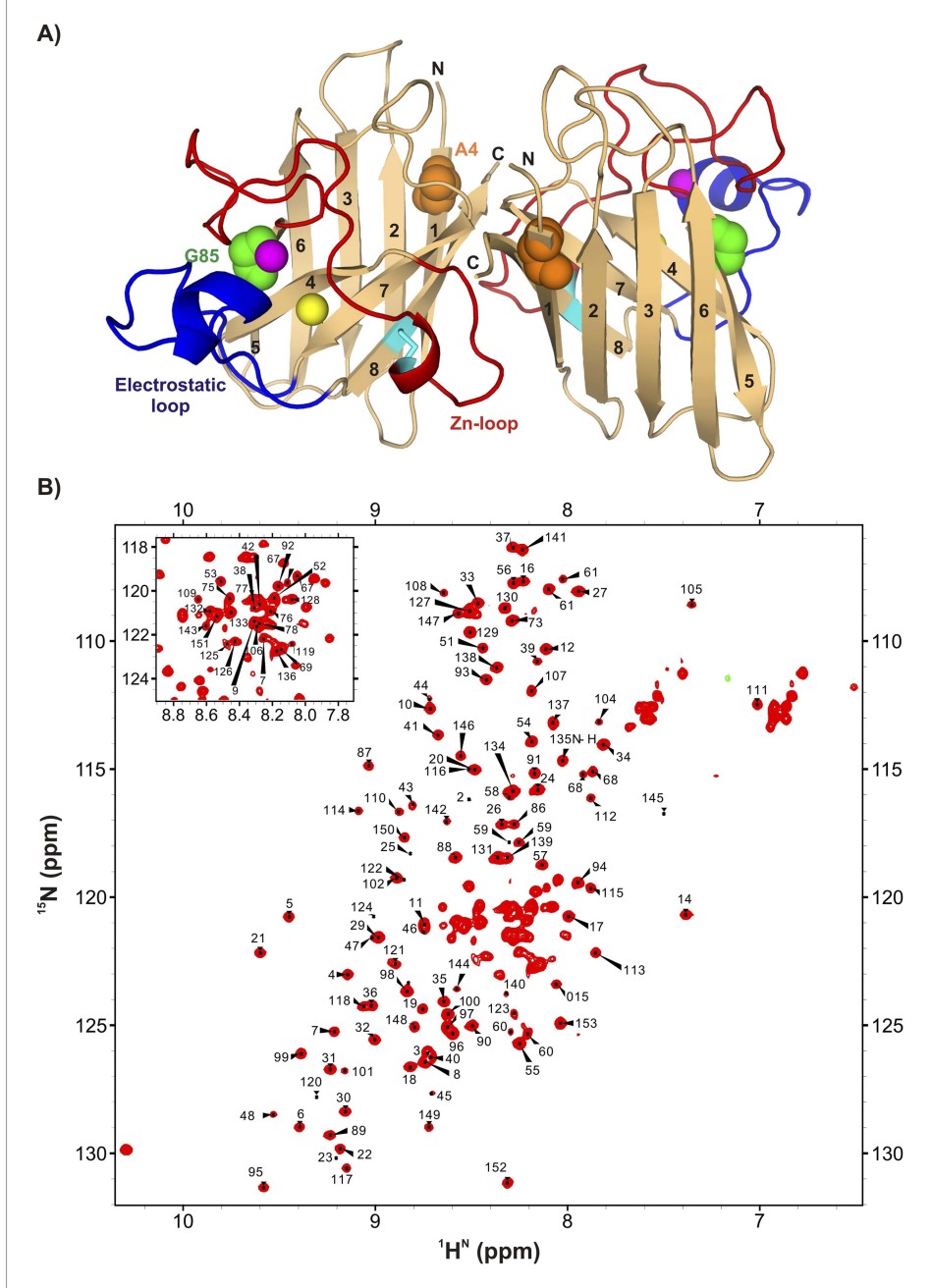

**Figure 1**. ApoSOD1[2SH] is folded in solution. (**A**) Structure of the mature homodimeric form of SOD1 (Cu$_2$Zn$_2$SOD1[S–S]) (pdb code: 1hl5) (*Strange et al., 2003*). Each monomer comprises an 8-stranded β-barrel with two long loops, the Zn-loop (red) and the electrostatic loop (blue), an intra-subunit disulfide bond (cyan), and one bound Cu (yellow) and Zn (magenta) ion. Each of the two long loops contains a short helix. Highlighted are positions 85 (green) and 4 (orange) that are sites of mutations discussed in the text. (**B**) $^1$H-$^{15}$N HSQC correlation map of pWT apoSOD1[2SH], 25°C (600 MHz $^1$H frequency). Chemical shift assignments of backbone amides are indicated on the plot.

## Transitions of pWT apoSOD1[2SH] leading to structural features found in Cu$_2$Zn$_2$SOD1[S–S]

In order to probe the energy landscape of pWT apoSOD1[2SH] we first carried out chemical exchange saturation transfer (CEST) experiments (*Forsén and Hoffman, 1963*; *Fawzi et al., 2011*;

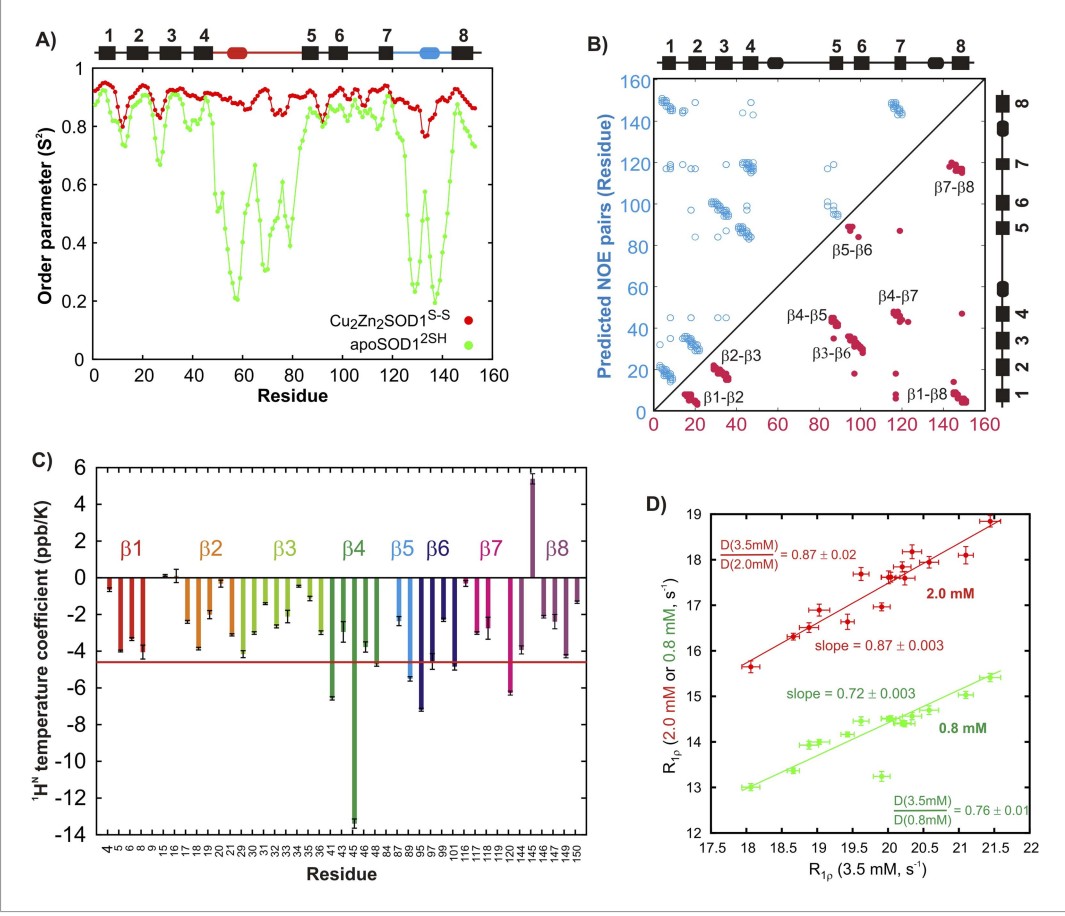

**Figure 2**. Structural features of the ground state of immature apoSOD1$^{2SH}$. (**A**) Chemical shift-derived order parameters squared (*Berjanskii and Wishart, 2005*), S$^2$, for the mature Cu$_2$Zn$_2$SOD1$^{S–S}$ (red; based on published chemical shifts [*Banci et al., 2002a*]) and the immature apoSOD1$^{2SH}$ (green) forms. Secondary structural elements of Cu$_2$Zn$_2$SOD1$^{S–S}$ are depicted above the plot, with the electrostatic loop shown in blue and the Zn-loop in red. (**B**) Correlation between inter-strand NOE pairs observed experimentally for apoSOD1$^{2SH}$ (red) and predicted based on the crystal structure of Cu$_2$Zn$_2$SOD1$^{S–S}$ (*Strange et al., 2003*) (blue). The secondary structure of Cu$_2$Zn$_2$SOD1$^{S–S}$ is indicated on the sides of the plot. (**C**) Temperature coefficients of amide protons of apoSOD1$^{2SH}$ known to be involved in inter-strand hydrogen bonds in the crystal structure of Cu$_2$Zn$_2$SOD1$^{S–S}$ (*Strange et al., 2003*). A cutoff of −4.6 ppb/K, often used to distinguish between protected and unprotected amides, is shown in red (*Cierpicki and Otlewski, 2001*). (**D**) Correlation between $^{15}$N R$_{1\rho}$ values of apoSOD1$^{2SH}$ at 3.5 mM (x-axis) and 2.0 mM (red) or 0.8 mM (green) (y-axis). Only $^{15}$N nuclei in ordered regions of apoSOD1$^{2SH}$ with R$_{ex}$ values smaller than 2 s$^{−1}$ in CPMG measurements (600 MHz, 3.5 mM protein concentration) were chosen for this analysis. Solid lines represent best fits of the data to an equation of the form y = mx + c and the slope in each case is indicated on the plot. Ratios of translational diffusion coefficients, D, of apoSOD1$^{2SH}$ at the indicated concentrations, that report on differences in sample viscosities with protein concentration, are also indicated.

*Lauzon et al., 2011*; *Vallurupalli et al., 2012*). In this class of relaxation experiment, a series of spectra are recorded whereby a weak radiofrequency (r.f.) perturbation is applied over a range of different frequencies (1 per spectrum) and the effects monitored by quantifying the intensity of ground state peaks. When the r.f. is positioned close to an invisible excited state resonance the effect of the perturbation will be transferred to the ground state peak via chemical exchange, leading to a decrease in the intensity of the observed ground state resonance. When the r.f. is applied near the resonance position of the ground state correlation there is attenuation as well, via a well-characterized saturation effect (*Carrington and McLachlan, 1967*). Thus, a plot of the intensity of the ground state peak as a function of the position of the r.f. field produces a CEST profile with a main dip at the

chemical shift of the resonance belonging to the major conformation and a smaller dip at the position of the corresponding resonance frequency of the minor conformer.

CEST experiments recorded on apoSOD1[2SH] show the presence of additional states with chemical shifts significantly different from the native conformation (*Figure 3* and *Figure 3—figure supplement 1*). In CEST profiles, the size of the minor dip is sensitive to both the fractional population of the excited state and its rate of exchange with the ground state (*Fawzi et al., 2011*; *Vallurupalli et al., 2012*). For exchange process I, the size of the minor dip increases with protein concentration (*Figure 3A* and *Figure 3—figure supplement 1*, green and red curves), implying an association or oligomerization event. Backbone

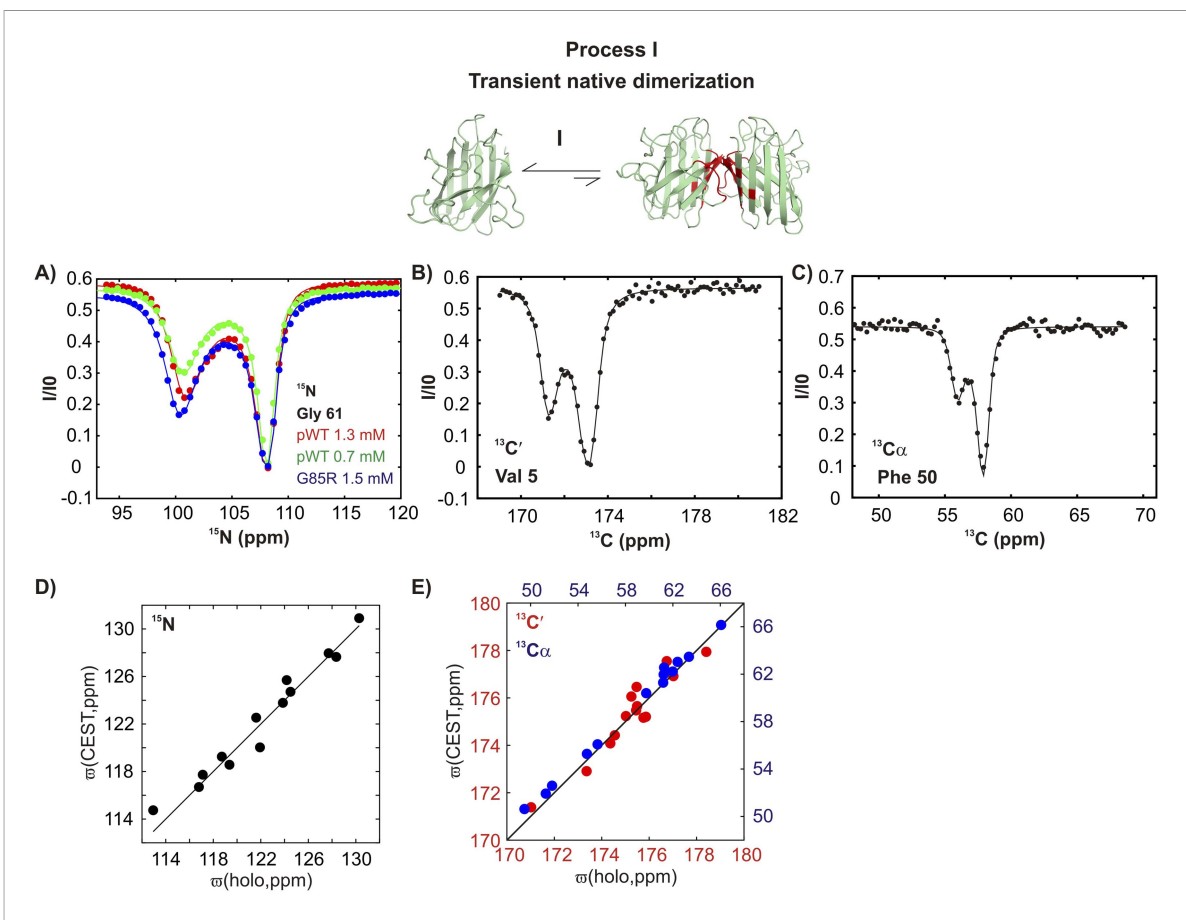

**Figure 3**. pWT apoSOD1[2SH] transiently samples a native dimer conformation. (**A**) [15N]-CEST profiles for G61 of pWT apoSOD1[2SH] that reports on the dimerization process, red: 1.3 mM, green: 0.7 mM and G85R apoSOD1[2SH] (blue: 1.5 mM). The G85R mutation does not interfere with transient native dimer formation as indicated by the large minor state dip. Profiles plot the intensity of the ground state peak (I) as a function of the position of the perturbing $B_1$ field (31 Hz), normalized to the corresponding intensity obtained from a spectrum recorded without the CEST element (I0) (*Vallurupalli et al., 2012*). [13C']- (**B**) and [13Cα]- (**C**) CEST profiles for residues V5 and F50, respectively. Correlations of [15N] (**D**), [13C'] (**E**, red) and [13Cα] (**E**, blue) CEST-derived excited state chemical shifts, ($\varpi$, ppm, y-axis) of apoSOD1[2SH] with the ground state shifts (*Banci et al., 2002a*) of $Cu_2Zn_2SOD1^{S–S}$ (x-axis) for residues localized to regions involved in transient dimerization.

The following figure supplements are available for figure 3:

**Figure supplement 1**. Conformationally excited states of apoSOD1[2SH] containing 'native' features as studied by [15N] (*Vallurupalli et al., 2012*), [13C'] (*Vallurupalli and Kay, 2013*), [13Cα] (*Long et al., 2014*), and [13CH3] (*Bouvignies et al., 2014*) CEST—process I.

**Figure supplement 2**. Estimating excited state lifetimes and populations for processes I and II.

**Figure supplement 3**. Excited state apoSOD1[2SH] conformations that are observed in ground state, mature $Cu_2Zn_2SOD1^{S–S}$ are generated via metal-independent processes.

$^{15}$N, $^{13}$C', and $^{13}$C$\alpha$ chemical shifts of the excited state involved in this process, as measured by the positions of minor state dips in CEST profiles (*Figure 3A–C* and *Figure 3—figure supplement 1*), localize to elements of structure forming the native dimer in Cu$_2$Zn$_2$SOD1$^{S–S}$ and correlate well with published chemical shifts of Cu$_2$Zn$_2$SOD1$^{S–S}$ (*Figure 3D,E*) (*Banci et al., 2002a*). This establishes that pWT apoSOD1$^{2SH}$ transiently forms native-like dimers.

In contrast, a second process, referred to as process II subsequently, is concentration-independent (*Figure 4A* and *Figure 4—figure supplement 1*), although $^{15}$N, $^{13}$C', and $^{13}$C$\alpha$ chemical shifts of the associated excited state (*Figure 4A–C*) also correlate well with those from Cu$_2$Zn$_2$SOD1$^{S–S}$ (*Figure 4D,E*). Residues belonging to process II are localized to a region in the electrostatic loop that is unstructured in apoSOD1$^{2SH}$ yet forms a short helix in Cu$_2$Zn$_2$SOD1$^{S–S}$ (*Valentine et al., 2005*). Process II is thus a transient local folding event involving formation of a helix from a disordered region in the ground state. This conclusion is further reinforced by the secondary structural propensity (SSP) score (*Marsh et al., 2006*) calculated from chemical shifts of the excited state for nuclei reporting on process II (residues 130–140). SSP scores for the ground and excited states computed from chemical

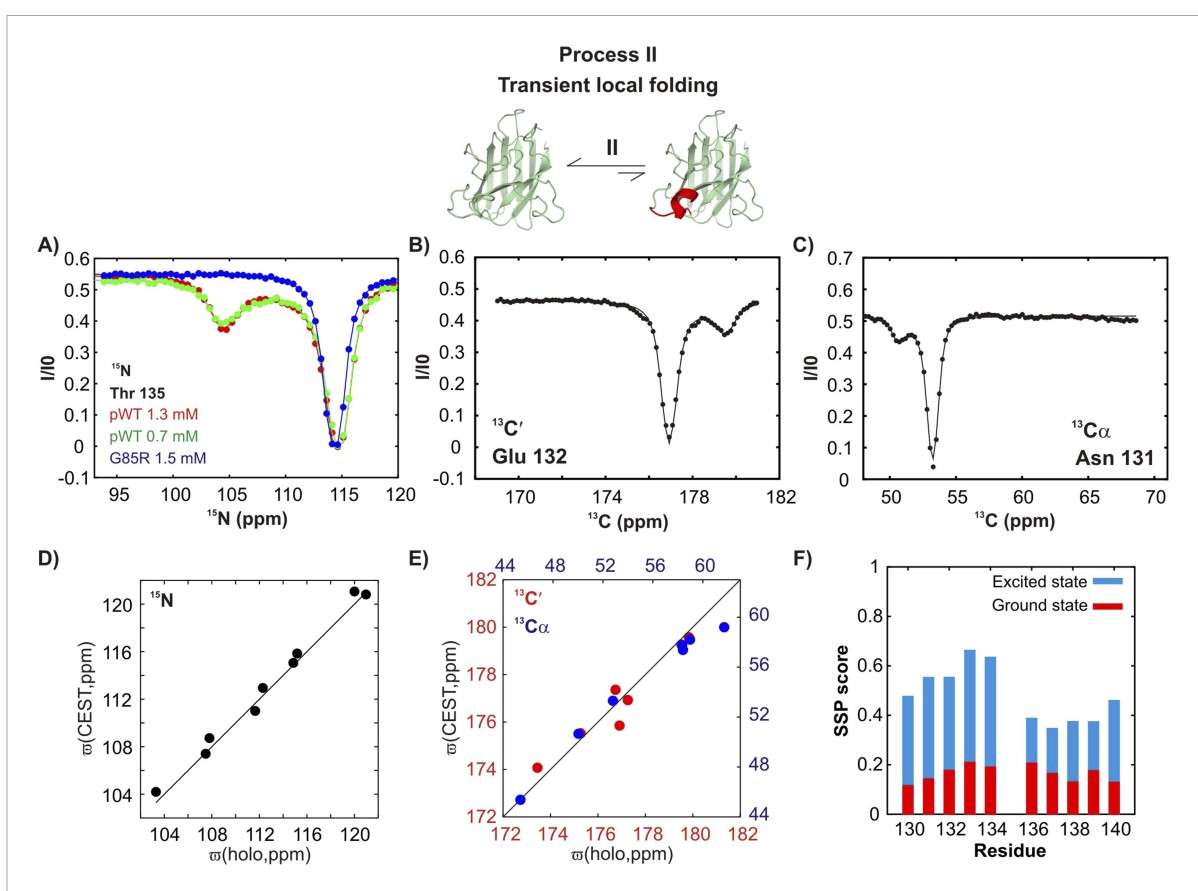

**Figure 4**. An excited state of pWT apoSOD1$^{2SH}$ with a native-like helix in the electrostatic loop. $^{15}$N- (**A**), $^{13}$C'- (**B**) and $^{13}$C$\alpha$- (**C**) CEST profiles for T135, E132 and N131 that report on transient helix folding. Correlations of $^{15}$N (**D**), $^{13}$C' (**E**, red) and $^{13}$C$\alpha$ (**E**, blue) CEST-derived excited state chemical shifts, ($\varpi$, ppm, y-axis) of apoSOD1$^{2SH}$ with the ground state shifts (*Banci et al., 2002a*) of Cu$_2$Zn$_2$SOD1$^{S–S}$ (x-axis) for residues localized to regions involved in transient helix formation. Helix formation is eliminated by the G85R mutation (**A**, blue). (**F**) Secondary Structural Propensity (SSP) (*Marsh et al., 2006*) scores computed from chemical shifts of nuclei in the ground and excited states and localized to residues 130–140 that are sensitive to process II. A value of 0 indicates a random coil conformation and a value of +1 corresponds to a fully formed helix. The helicity increases by up to ~50% in the excited state, showing that process II involves local folding.

The following figure supplement is available for figure 4:

**Figure supplement 1**. Conformationally excited states of apoSOD1$^{2SH}$ containing 'native' features as studied by $^{15}$N (*Vallurupalli et al., 2012*), $^{13}$C' (*Vallurupalli and Kay, 2013*), $^{13}$C$\alpha$ (*Long et al., 2014*), and $^{13}$CH$_3$ (*Bouvignies et al., 2014*) CEST—process II.

shifts of the two interchanging conformers show that the helical content in this region of apoSOD1$^{2SH}$ increases considerably (by up to ~50%) in the excited state, establishing that process II involves local structuring of a helix (*Figure 4F*). Notably both processes I and II are independent of each other since the ALS mutant G85R (*Cao et al., 2008*) eliminates II with little effect on I (*Figures 3A, 4A* and *Figure 3—figure supplement 1*, *Figure 4—figure supplement 1*) and both involve the partial transformation of apoSOD1$^{2SH}$ into a mature native-like state, limiting the flexibility of the Zn-binding and electrostatic loops (processes I, II respectively) and sequestering key structural elements, such as strands β1 and β8 (process I). Fits of the processes (*Figure 3—figure supplement 2*), as described in 'Materials and methods', provide an estimate of the population and lifetime of each excited state (25℃), with values of 2%, 13 ms and 3%, 3 ms for the helix and the native dimer respectively at an overall protein concentration of 1.3 mM, along with a dimer dissociation constant of 85 ± 50 mM.

Both local folding and transient dimerization are metal-independent processes (*Figure 3—figure supplement 3*) that result in conformations observed in mature Cu$_2$Zn$_2$SOD1$^{S–S}$. Thus the amino acid sequence can code for the organization of the electrostatic and the Zn-binding loops in a manner independent of metal binding. Our results establish that metal binding is not a requirement for loop organization and further that zinc binding may, in part, proceed through a conformational selection mechanism whereby zinc binds to excited state conformers that contain the transiently formed helix in the electrostatic loop. The negative end of this helix dipole, which has been postulated to stabilize the metal-bound conformation in SOD1 (*Cao et al., 2008*), may also facilitate zinc ligation by guiding the metal to the binding pocket through electrostatic interactions. In this regard, it is noteworthy that the G85R mutant, which has a reduced affinity for zinc (*Valentine et al., 2005*), does not transiently form this helix as established by the absence of minor state CEST dips for the helix forming residues (*Figure 4A* blue, *Figure 4—figure supplement 1*).

## Transitions of pWT apoSOD1$^{2SH}$ leading to aberrant oligomers

Additional millisecond timescale processes have been characterized using Carr-Purcell-Meiboom-Gill (CPMG) relaxation dispersion NMR (*Carr and Purcell, 1954*; *Meiboom and Gill, 1958*; *Palmer et al., 2000*) that takes advantage of the increase in effective transverse relaxation rates of NMR spins (R$_{2,eff}$) resulting from stochastic fluctuations between protein conformations. In this experiment, transverse relaxation rates of NMR probes are modulated by applying a variable number of chemical shift refocusing pulses during a fixed time element (*Mulder et al., 2001*). Application of increasing numbers of pulses decreases the effective chemical shift differences between exchanging sites, resulting in a decrease in the measured effective transverse relaxation rate. Thus, dispersion profiles, R$_{2,eff}$ vs rate of application of refocusing pulses ($\nu_{CPMG}$), showing a decrease in R$_{2,eff}$ with increasing $\nu_{CPMG}$ are a hallmark of chemical exchange, while flat profiles indicate either that there is no exchange or that it proceeds much faster than the rate of application of the pulses.

$^{15}$N and $^{13}$CH$_3$ CPMG experiments (*Lundström et al., 2007b*; *Vallurupalli et al., 2007*) on apoSOD1$^{2SH}$ allow characterization of two additional millisecond timescale processes, referred to as processes III and IV (*Figures 5A, 6A*). The positions of $^{15}$N and methyl-$^{13}$C spins identified as probes of processes III and IV are highlighted in monomers of SOD1 in *Figures 5B, 6B*, respectively. Both processes are concentration dependent (*Figures 5C, 6C* and *Figure 5—figure supplement 1*), as seen from the significant increase in contributions to transverse relaxation rates from chemical exchange, R$_{ex}$, with increasing total protein concentration, and hence correspond to transient oligomer formation. These processes can be separated on the basis of their temperature dependencies with R$_{ex}$ values from III and IV increasing and decreasing, respectively, with increasing temperature (*Figures 5D, 6D* and *Figure 5—figure supplement 1*). In order to quantify the differences in kinetics and thermodynamics of these exchange events in more detail we have further analyzed the dispersion data assuming a simple monomer (M)—dimer (D) equilibrium model (see 'Materials and methods', 'Separating probes of processes I–IV' for details), yielding excited state populations and lifetimes of 3.3 ± 0.2%, 6.0 ± 0.6 ms and 2.1 ± 0.1%, 1.6 ± 0.1 ms for processes III and IV, respectively, at 25℃, 1.3 mM protein. The populations and lifetimes of all of the apoSOD1$^{2SH}$ states characterized in the present study are listed in *Table 1*.

*Figure 7A–C* shows distinct temperature dependencies for the dimerization equilibrium constant, K$_{MD}$, and for the underlying rate constants k$_{MD}$, k$_{DM}$ for each process, further highlighting that exchange events III and IV are independent. Indeed, all 3 oligomerization processes are independent of each other, as CPMG profiles reflecting the native dimerization process (I) cannot be well-fit using

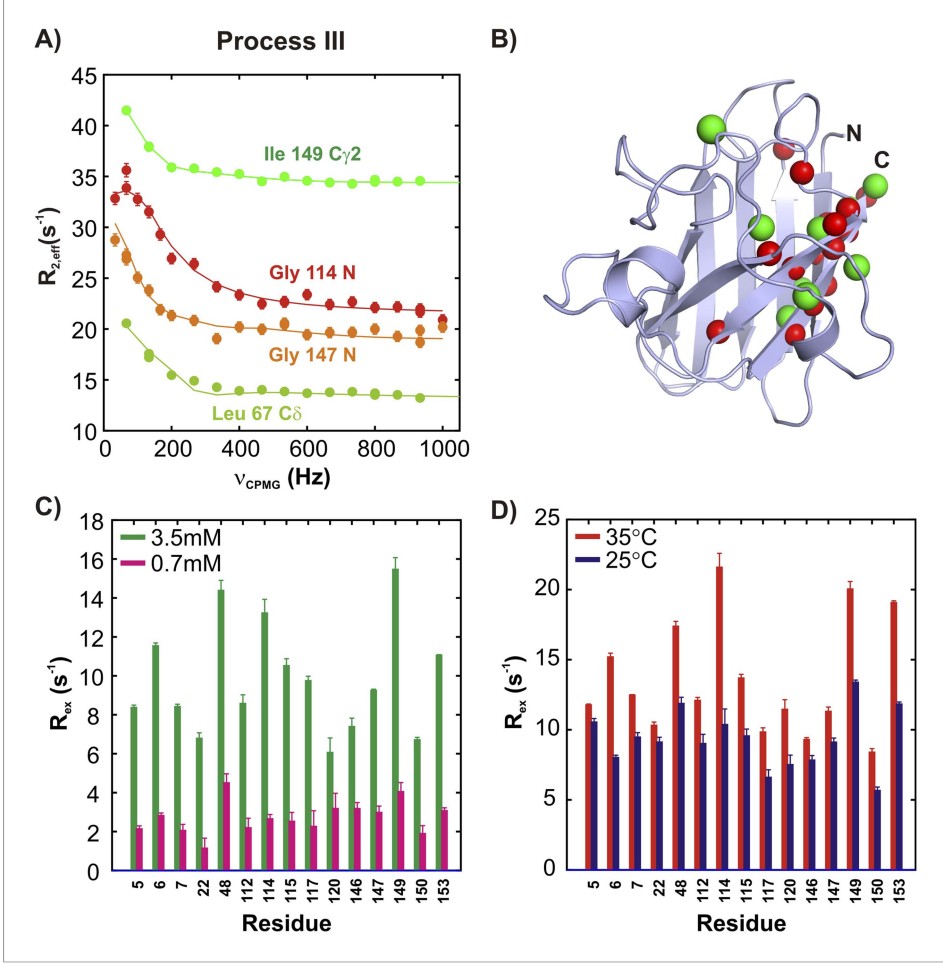

**Figure 5**. ApoSOD1²SH transiently forms distinct non-native oligomers—Process III. Representative ¹⁵N (red, orange) and ¹³CH₃ (green) CPMG profiles for residues belonging to processes III (**A**). Each curve shows the effective ¹⁵N or ¹³C transverse relaxation rate, R₂,ₑff, plotted as a function of the rate of pulsing in CPMG trains (*Palmer et al., 2000*). (**B**) Residues identified as reporting on process III via CPMG experiments (see 'Materials and methods') are highlighted on the structure of a monomer of Cu₂Zn₂SOD1ˢ⁻ˢ. Concentration (**C**, 25°C) and temperature-dependence (**D**, 2 mM protein concentration) of Rₑₓ values (the difference in relaxation rates recorded at the lowest and highest CPMG field strengths [*Palmer et al., 2000*]) obtained from ¹⁵N CPMG measurements on pWT apoSOD1²SH.

The following figure supplement is available for figure 5:

**Figure supplement 1**. The two non-native oligomerization processes (III and IV) are distinct from one another.

exchange parameters from analysis of process III ('Materials and methods' 'Separating probes of processes I–IV' and *Figure 7—figure supplement 1*), while processes I and IV have distinct temperature dependencies (*Figure 7—figure supplement 1*). Thus, exchange processes III and IV correspond to the formation of non-native oligomers.

## Complications arising from multiple exchange processes

The presence of at least four distinct exchange processes in apoSOD1²SH suggests that the underlying free energy landscape of this protein is rugged. Further, the pervasive nature of exchange in this system significantly complicates analysis of the CEST and CPMG data. This is especially the case because there is considerable overlap in the regions that report on the various processes, with residues associated with the native dimer interface participating in three of the four exchange events

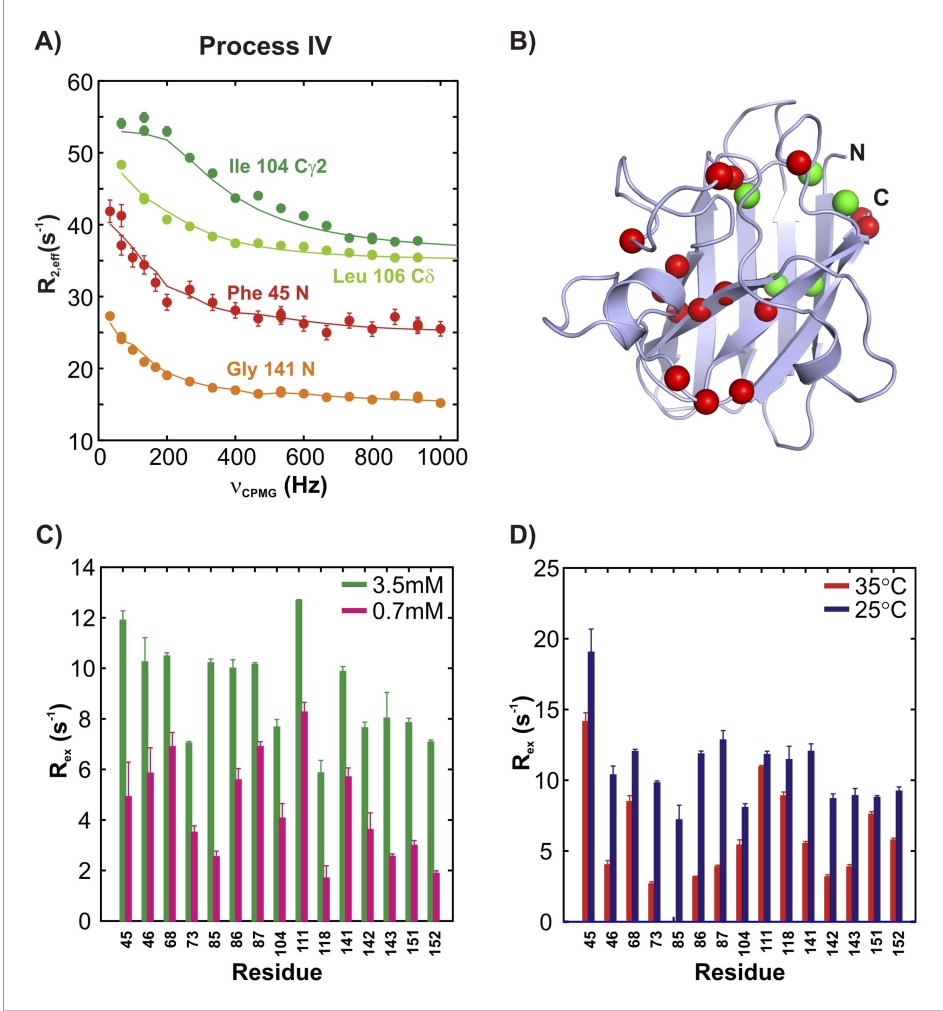

**Figure 6**. ApoSOD1$^{2SH}$ transiently forms distinct non-native oligomers—Process IV. (**A**–**D**) As **Figure 5** with the exception that the residues selected are those probing process IV.

that have been characterized. Not surprisingly, therefore, in some cases the relaxation data show complex exchange behaviour. This does not necessarily imply that the four exchange events discussed here are coupled, but can also result from spin probes reporting on one process 'sensing' fluctuating fields from proximal exchange events. For example, CEST profiles of G51, N53, T54, and G61 clearly establish that these residues report on native dimerization, with shift differences in excess of ~4 ppm between ground and excited state conformers. Yet, for these residues, $^{15}$N CPMG relaxation dispersion profiles recorded on pWT samples could not be fit to models of two-site exchange. In contrast, the corresponding profiles from a G85R apoSOD1$^{2SH}$ mutant, where process II is eliminated and process IV reduced, could be fit to the simplest two-state exchange model. Further, the CEST profiles for these residues in pWT and G85R apoSOD1$^{2SH}$ are very similar, suggesting that process I has not been significantly affected via the mutation. A likely explanation is that the 'seemingly more simple exchange behaviour' in the G85R mutant is the result of the elimination of local field fluctuations from these additional processes. The inability to fit the dispersion data for these residues in the pWT protein is, thus, at least in part the result of fluctuating local magnetic fields that arise from a number of simultaneous proximal processes.

Most certainly, the multiple exchange processes do complicate assignment of residues to distinct exchange events. A detailed description of the approach taken, involving studies at a number of different temperatures and protein concentrations, is provided in 'Materials and methods' 'Separating

**Table 1**. Excited state populations ($p_E$) and lifetimes ($\tau_E$) for processes I–IV

| Process | I | II | III | IV |
|---|---|---|---|---|
| $p_E$ (%) | 3 ± 1 | 2.1 ± 0.1 | 3.3 ± 0.2 | 2.1 ± 0.1 |
| $\tau_E$ (ms) | 3.0 ± 0.6 | 13 ± 1 | 6.0 ± 0.6 | 1.6 ± 0.1 |

Uncertainties for process I are calculated as ±1 s.d. of the values obtained from single $B_1$ field fits of $^{15}N$ CEST profiles of G51, N53, T54, and G61 (**Figure 3—figure supplement 2B**). Values for process II are the mean and standard deviation of $p_E$ and $\tau_E$ obtained from the two best three-state models (**Figure 3—figure supplement 2E**). The equations for calculating these values are listed in the legend for **Figure 3—figure supplement 2**. Note that very similar values of $p_E$ and $\tau_E$ are obtained from the two models (on- and off-pathway) and it is not possible to distinguish between them on the basis of the NMR data. Uncertainties for exchange parameters describing processes III and IV are obtained from global two-state fits of CPMG profiles reporting on each process. All values are relevant for a 1.3 mM protein concentration, 25°C.

processes I–IV'. **Figure 8** and **Table 2** summarize the criteria used for separating residues into distinct exchange processes and **Figure 8** further indicates what residues were subsequently used as constraints in structure calculations (see below). While not all residues reporting on each exchange process could be identified, a sufficient number of probes could be selected for each process both to identify its nature and to enable the structural characterization of the resulting excited state conformer (as described below).

## Structural models of excited states of pWT apoSOD1$^{2SH}$

Having established that apoSOD1$^{2SH}$ transiently accesses at least two non-native oligomeric conformations, we next obtained structural models for these conformers by assuming that they are dimeric (see 'Materials and methods' 'Choice of dimer models for the association process') and by using excited state chemical shifts as restraints in the biomolecular docking program HADDOCK (**Dominguez et al., 2003**). Residues that were used as restraints in structure calculations are a subset of those reporting on each process as outlined in **Figure 8**. They are listed in **Table 3** and their positions in the structure of SOD1 are depicted in **Figure 9—figure supplement 1A**. As a first step and in order to validate the methodology the structure of the transient native-like dimer formed via process I was determined (**Supplementary File 1**) and found to be superimposable with the dimeric holo-SOD crystal structure (**Valentine et al., 2005**) with a pair-wise all-atom RMSD of 0.9 Å from the crystal structure to the lowest energy HADDOCK derived model (**Figure 9A**).

Using similar HADDOCK calculations structural models were generated for the non-native dimers as well (**Figure 9—figure supplement 1**, **Supplementary Files 2** and **3**, see 'Materials and methods' 'Building structural models of excited states using HADDOCK'). These differ significantly from the

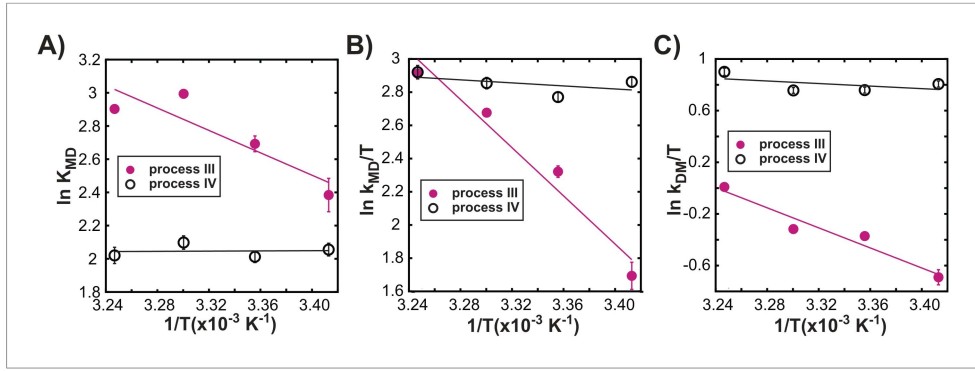

**Figure 7**. Processes III and IV are distinct. Variation in the logarithms of the equilibrium constant for dimer formation (**A**, $K_{MD}$), as well as the association (**B**, $k_{MD}$) and dissociation (**C**, $k_{DM}$) rate constants with inverse temperature for processes III (closed magenta circles) and IV (open black circles) obtained as described in 'Materials and methods', 'Separating probes of processes I–IV'.

The following figure supplement is available for figure 7:

**Figure supplement 1**. The two transient non-native oligomerization processes (III and IV) observed in apoSOD1$^{2SH}$ are distinct from native dimerization (process I).

## Summary of criteria for assigning residues to processes

### Process I

CEST-based $^{15}$N $\Delta\varpi$ values of G51 (8.3 ppm), N53 (8.1 ppm), T54 (3.8 ppm) and G61 (-8 ppm) all agree well with those expected for native dimerization based on chemical shifts of holo SOD

These residues
- localize to the native dimer interface
- have CEST profiles that are concentration-dependent (increase with increasing concentration) (Figure 3, Figure 3-figure supplement 1)
- have $R_{ex}$ values in CPMG dispersions that increase with temperature (Figure 7-figure supplement 1)
- have $R_{ex} < 3$ s$^{-1}$ in the mutant G85R, that is the maximum that can be expected based on the slow exchange limit CPMG profiles of G51, N53, T54 and G61 (see Materials and Methods, Figure 7-figure supplement 1)

**Residue selection**
Residues were selected as reporting on process I if they bury > 2 A$^2$ surface area upon native dimerization based on the crystal structure of holo SOD (20 residues) (Table 3)

**Restraint selection for HADDOCK**
Residues selected above for which $\Delta\varpi$ > 0.3 ppm as well as a SASA* value in the monomer > 40% (14 residues) (Table 3)

### Process II

CEST-based $^{15}$N $\Delta\varpi$ of T135 (-10.5 ppm) and T137 (-8.9 ppm) agree well with values expected for helix formation based on chemical shifts of holo SOD

These residues
- have CEST profiles that are concentration-independent (Figure 4, Figure4-figure supplement 1)

**Residue selection**
Residues 130-140 in the electrostatic loop were selected as reporting on process II (11 residues) (Table 3)

These residues were used to calculate SSP scores to establish the presence of increased helical structure in this region

### Process III

Residue selection based on $^{15}$N and $^{13}$CH$_3$ CPMG

These residues
- have $R_{ex}$ values larger than 5 s$^{-1}$ (3.5 mM ($^{15}$N) or 2.5 mM ($^{13}$CH$_3$), 25°C) (Figure 5, Figure 5-figure supplement 1)
- have $R_{ex}$ values that are concentration-dependent (increase with increasing concentration) (Figure 5, Figure 5-figure supplement 1)
- have $R_{ex}$ values that increase with temperature (Figure 5, Figure 5-figure supplement 1)
- have $R_{ex} > 3$ s$^{-1}$ in the mutant G85R (see Materials and Methods, Figure 7-figure supplement 1) that distinguishes them from process I residues
- have $^{15}$N and $^{13}$CH$_3$ CPMG profiles that can be globally fit to a two-state model (Figure 5)

**Residue selection**
Residues were selected as reporting on process III if they satisfied all criteria above (23 residues) (Table 3)

**Restraint selection for HADDOCK**
Residues selected above which also have a SASA* value in the monomer > 40% (7 residues) (Table 3)

### Process IV

Residue selection based on $^{15}$N and $^{13}$CH$_3$ CPMG

These residues
- have $R_{ex}$ values larger than 5 s$^{-1}$ (3.5 mM ($^{15}$N) or 2.5 mM ($^{13}$CH$_3$), 25°C) (Figure 6, Figure 5-figure supplement 1)
- have $R_{ex}$ values that are concentration-dependent (increase with increasing concentration) (Figure 6, Figure 5-figure supplement 1)
- have $R_{ex}$ values that decrease with temperature (Figure 6, Figure 5-figure supplement 1)
- have $^{15}$N and $^{13}$CH$_3$ CPMG profiles that can be globally fit to a two-state model

**Residue selection**
Residues were selected as reporting on process IV if they satisfied all criteria above (19 residues) (Table 3)

**Restraint selection for HADDOCK**
Residues selected above which also have a SASA* value in the monomer > 40% (7 residues) (Table 3)

* SASA is the total backbone or side-chain solvent accessible surface area

**Figure 8**. Summary of the criteria used for assigning exchange processes I–IV. More details can be found in 'Materials and methods', 'Separating probes of processes I–IV'.

mature dimeric form of Cu$_2$Zn$_2$SOD1$^{S–S}$ (**Figure 9B**). The non-native dimer formed via process III (referred to as non-native dimer 1) is symmetric and topologically related to the native dimer by a 180° rotation of one of the monomer units about an axis shown in **Figure 9B**. It is stabilized by an interface

**Table 2.** Distinguishing between exchange processes I–IV

|  | I (native dimer) | II (local folding) | III (non-native dimer 1) | IV (non-native dimer 2) |
|---|---|---|---|---|
| I (Native dimer) | – | – | – | – |
| II (Local folding) | Concentration dependence* | – | – | – |
| III (Non-native dimer 1) | $R_{ex}$ values in G85R and $p_E/\tau_E$ values from G85R† | Concentration dependence§ | – | – |
| IV (Non-native dimer 2) | Temperature dependence of CPMG $R_{ex}$ values‡ | Concentration dependence# | Temperature dependence of CPMG $R_{ex}$ values¶ | – |

*__Figure 3__, __Figure 3—figure supplement 1__, __Figure 4__ and __Figure 4—figure supplement 1__.
†__Figure 7—figure supplement 1__.
‡__Figure 7—figure supplement 1__.
§__Figure 4__, __Figure 4—figure supplement 1__, __Figure 5__, __Figure 5—figure supplement 1__.
#__Figure 4__, __Figure 4—figure supplement 1__, __Figure 6__, __Figure 5—figure supplement 1__.
¶__Figure 5__, __Figure 6__, __Figure 5—figure supplement 1__.

very similar to that of the native dimer, involving Zn-binding and β6-β7 loops, as well as strands β1 and β8. Notably, the restraints used to model the excited state formed via this process (__Table 3__ and __Figure 9—figure supplement 1__) localize to the native dimer interface on one side of the monomer, consistent with the symmetric dimer structural model. These restraints are a subset of those for process I, and it is reasonable to ask why the native structure is not generated in this case. This can be understood by noting that the native dimer interface involves contacts between residues from β1–β8 in one monomer and residues localized to the Zn-loop from a second, placing the corresponding β1–β8 residues in the second monomer far from those in the first. In contrast, since the restraint residues for process III include those from β1–β8 but not the Zn-loop, the process of satisfying the restraints (i.e., bringing restraint residues within a prescribed distance from each other) places β1–β8 from both monomers of the dimer in proximity, leading to a structure where one of the monomers is rotated 180° compared to the native dimer.

Unlike both dimeric structures formed via processes I and III, the second non-native dimer (non-native dimer 2, process IV) is asymmetric, with the orientation of the monomers with respect to each other illustrated in __Figure 9B__. In this case the interface is formed by the β6–β7 and Zn loops as well as strands 1 and 8 from one monomer and by the electrostatic and Zn-binding loops from the second molecule. The asymmetry is to be expected based on the location of the input restraints (__Figure 9—figure supplement 1__), which are not localized to a particular region of the molecule, but spread into a number of clusters. These clusters of restraints cannot be simultaneously satisfied by a symmetric dimer, but can be readily accounted for by an asymmetric dimeric structure that brings one cluster on one monomer close to the remaining clusters on a second molecule.

The structural models all point to an important role for strand β1 in the formation of excited states derived from processes I, III, and IV. This leads to the prediction that mutations in this region could disrupt association. To test this, we mutated A4 in β1 to V and recorded further CPMG experiments to evaluate its effects on the association processes. As expected, the effect is pronounced with both native (I) and non-native dimerization processes (III) completely abolished, while formation of the asymmetric dimer via process IV is considerably diminished (__Figure 9C__).

## Discussion

In this study, we have used a combination of CEST and CPMG NMR relaxation experiments to probe the energy landscape of apoSOD1$^{2SH}$, an immature form of SOD1 that is monomeric, metal free and lacking the stabilizing disulfide bond between C57 and C146. In the absence of metal and the disulfide bond, backbone amides of both the Zn-binding and electrostatic loops have much smaller order parameters than those of $Cu_2Zn_2SOD1^{S-S}$ (__Figure 2__). The increased dynamics and corresponding lack of defined structure likely play a significant role in the promiscuous interactions that these loops make, leading to the formation of both native-like and non-native interactions.

We have shown that apoSOD1$^{2SH}$ converts to four different conformations from the monomeric ground state via separate chemical exchange events labeled processes I–IV and summarized in

**Table 3.** Residues reporting on processes I–IV†

| Process I‡ | Process II‡ | Process III§ | Process IV§ |
|---|---|---|---|
| V5* | G130 | V5 | A6 |
| **V7** | N131* | A6 | I18 |
| **K9** | E132* | **V7** | F45 |
| I17 | E133* | L8 | V46 |
| E49 | S134* | **I17** | **S68** |
| **F50*** | T135* | Q22 | G73 |
| **G51*** | K136* | V47 | G85 |
| **D52*** | T137* | H48 | **N86** |
| **N53*** | G138 | A60 | V87 |
| **T54*** | N139* | **L67** | I104 |
| **G61*** | A140* | I112 | L106 |
| **I113** | – | **G114** | S111 |
| **G114** | – | R115 | **I113** |
| R115 | – | L117 | V118 |
| **V148** | – | H120 | **G141** |
| I149* | – | C146 | **S142** |
| **G150*** | – | G147 | **R143** |
| **I151*** | – | **V148** | I151 |
| A152* | – | I149 | A152 |
| **Q153** | – | **G150** | – |
| – | – | I151 | – |
| – | – | A152 | – |
| – | – | **Q153** | – |

†Residues selected according to the criteria outlined in 'Materials and methods' 'Separating probes of processes I–IV'. Active residues used in the molecular docking program HADDOCK (***Dominguez et al., 2003***) are indicated in bold. A total of 28 and 14 restraints were used in structure calculations of symmetric dimers corresponding to excited states derived from processes I and III, respectively (these numbers are doubled to indicate that a restraint from monomer A to B also pertains from B to A), with 7 restraints for the calculation of the excited state from process IV (asymmetric). The energy funnels obtained in the structure calculations (***Figure 9—figure supplement 1***) support the view that the structures have converged. For the non-native dimers, we have further evaluated the robustness of the calculations by repeating them after first removing 1 restraint at a time. Very similar ensembles to those illustrated in ***Figure 9—figure supplement 1*** were obtained indicating that the structure calculation is not driven merely by a single restraint. For process III, we also obtain very similar structures if only restraints from $^{15}$N CPMG data are included. Moreover, we have run additional calculations for process IV starting with the NMR derived structure of monomeric C6A/C111S/Q133E/F50E/G51E apoSOD1$^{S–S}$ (PDB accession code 1RK7 [***Banci et al., 2003***]), in which the electrostatic and Zn-binding loops are disordered, rather than from a monomer of the native dimer X-ray structure (PDB accession code 1HL5 [***Strange et al., 2003***]). Very similar structures are obtained in all cases.
‡Residues labeled with * are those for which separate dips or shoulders corresponding to an excited state were observed in $^{15}$N, $^{13}$C' or $^{13}$C$^\alpha$ CEST profiles. Recall that relatively small $\Delta\varpi$ values precluded the use of CEST for the study of processes III and IV. Note that a number of active residues are classified as reporting on both processes I and III; these correspond to residues that are at the native dimer interface (and hence classified as reporters of process I, see 'Separating probes of processes I–IV, Process I: Transient native-like dimerization') and that have dispersion profiles and $R_{ex}$ values that are clearly sensitive to process III.
§$R_{ex}$ > 5 s$^{-1}$ (800 MHz) for $^{15}$N (3.5 mM) and/or $^{13}$C-methyl (2.5 mM) CPMG relaxation dispersion profiles.

***Figure 9B***. Some of the excited state conformers are decidedly native-like with one of the excited states corresponding to the Cu$_2$Zn$_2$SOD1$^{S–S}$ dimeric structure. Other exchange events lead to aberrant structures that are stabilized via interfaces comprised of many of the same residues that are

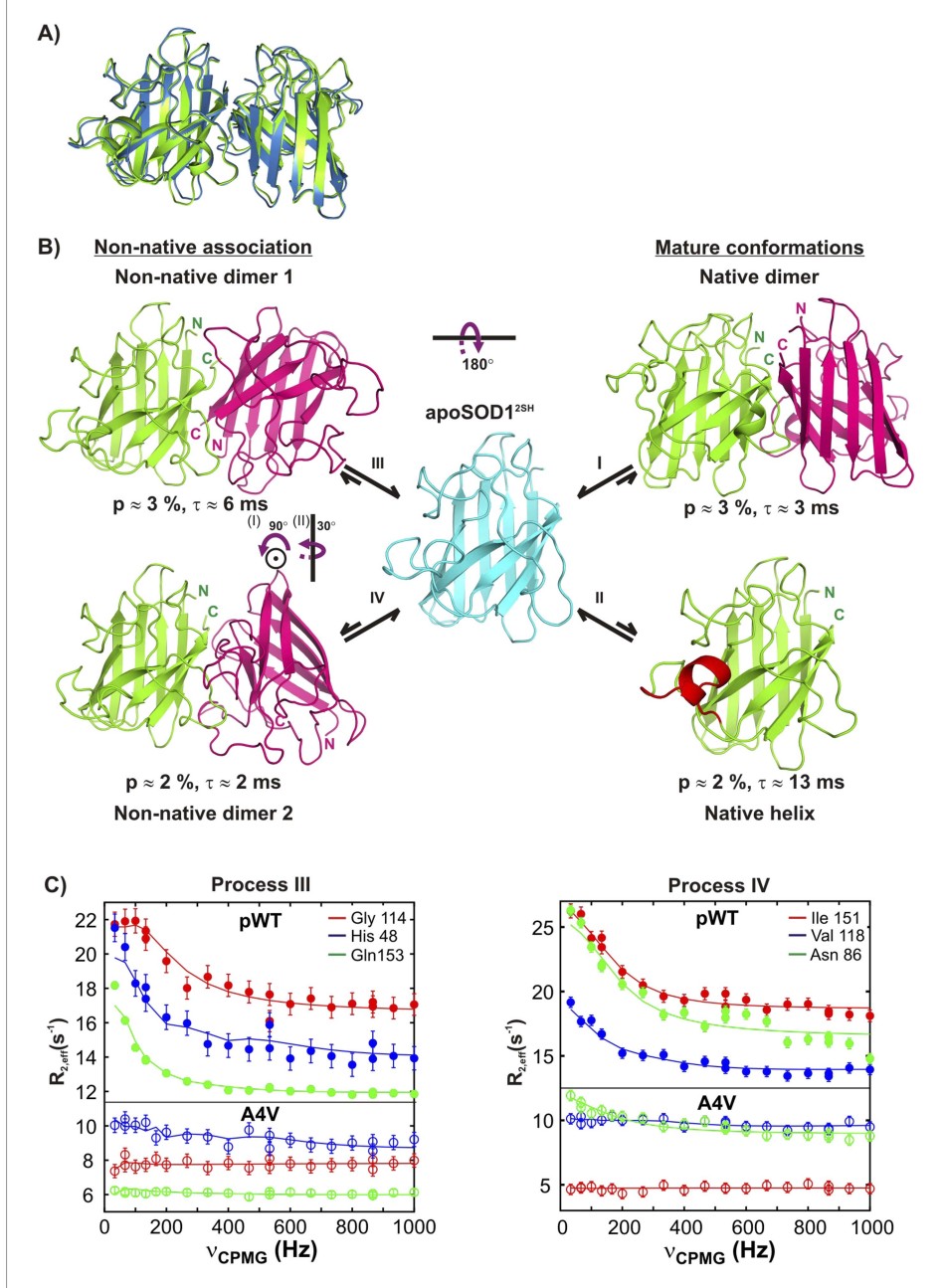

**Figure 9**. Monomeric apoSOD1$^{2SH}$ is in equilibrium with mature and aberrantly associated conformations. (**A**) Superposition of ribbon diagrams of the lowest energy HADDOCK model of the native dimer excited state of pWT apoSOD1$^{2SH}$ (process I, blue) and the X-ray structure of Cu$_2$Zn$_2$SOD1$^{S–S}$ (pdb code: 1hl5, green) (**Strange et al., 2003**). (**B**) Conformations transiently populated by apoSOD1$^{2SH}$ (blue), including states with structural features similar to the ground state of Cu$_2$Zn$_2$SOD1$^{S–S}$ ('Mature conformations') and aberrant oligomers ('Non-native association'). Structural models for the native dimer (I) and the symmetric (III) and asymmetric (IV) non-native dimers populated by apoSOD1$^{2SH}$ as determined from the HADDOCK program (**Dominguez et al., 2003**) are shown along with the model that includes helix folding (II) taken directly as a monomer from the crystal structure of Cu$_2$Zn$_2$SOD1$^{S–S}$ (**Strange et al., 2003**). The purple arrows indicate approximate rotations needed to transform magenta monomers of non-native dimer 1 to corresponding monomers of either the native dimer or non-native dimer 2. In the latter case, two successive rotations about (i) an axis perpendicular to the plane of the paper and subsequently about (ii) a vertical axis are required. Note that the scheme shown here is the simplest one consistent with the spin relaxation data (see 'Discussion'). (**C**) The A4V mutant severely reduces association via processes III and IV, as established by the significant decrease in the sizes of CPMG dispersion profiles. The top part of each panel

*Figure 9. Continued*

shows CPMG profiles for pWT apoSOD1$^{2SH}$, with the corresponding curves for the A4V mutant indicated in the bottom. A small constant y-offset has been applied to some of the curves for clarity.
The following figure supplements are available for figure 9:

**Figure supplement 1**. Structural models for the oligomeric excited states of pWT apoSOD1$^{2SH}$ from the biomolecular docking program HADDOCK (*Dominguez et al., 2003*).

**Figure supplement 2**. The asymmetric dimer from process IV can serve as a hub for the formation of larger oligomers.

involved in inter-molecular interactions in the native dimer (*Figure 10*). The presence of at least four distinct exchange processes is different from what has been reported previously based on CPMG studies of an oxidized variant of apoSOD1 containing a pair of dimer-destabilizing mutations in which F50 and G51 were substituted to Glu. There a single exchange event could be characterized but the molecular details of that process remain to be elucidated (*Teilum et al., 2009*).

Central to the generation of the structural models of thermally accessible excited states of pWT apoSOD1$^{2SH}$ (*Figure 9B*) has been the separation of residues belonging to the different exchange processes. Most applications of either CEST or CPMG relaxation experiments have involved systems where the majority of residues report on a single exchange event, although in some cases the processes have been more than two-state (*Korzhnev et al., 2004*; *Grey et al., 2006*; *Neudecker et al., 2006*; *Korzhnev et al., 2007*; *Sugase et al., 2007*). When more than a single exchange event is present residues reporting on a specific process may be localized structurally (*Vallurupalli and Kay, 2006*), simplifying the analysis. The situation is more complex for apoSOD1$^{2SH}$. Because the dimer interface plays critical roles in both native and non-native oligomer formation, residues do not cluster spatially to separate processes and a large number of different experiments are thus necessary to resolve each exchange event. Untangling the four exchange processes described in the present study has required a combination of both CEST and CPMG relaxation experiments, as described in 'Materials and methods' 'Separating probes of processes I–IV', and summarized in *Figure 8*. For example, an initial $^{15}$N CEST study showed large minor state dips for G51, N53, T54, and G61 that were very well separated from the ground state (8.3 ppm, 8.1 ppm, 3.8 ppm, and −7.3 ppm, respectively). These chemical shifts matched those of Cu$_2$Zn$_2$SOD1$^{S–S}$, so that a straightforward assignment of reporters of process I was obtained, along with the identification of the nature of the exchange event. By comparison, the corresponding CPMG dispersion profiles for these residues could not be well fit together, although those for N53 and T54 could be fit, that reflects the sensitivity of the dispersion profiles to more than a single exchange process. CEST profiles can, of course, also be sensitive to multiple exchange events and indeed, those from G51, N53, T54, and G61 are. However, unlike dispersion data that must be rigorously fit to extract shift differences, the chemical shifts of the excited state can be obtained by simple inspection of CEST profiles so long as they are significantly different from those of the ground state. In cases involving multiple processes where only one has large chemical shift differences between interconverting conformers it is therefore possible to get accurate shift values for residues of this excited state from the position of the minor dips, as is made clear from *Figures 3D,E, 4D,E*. In contrast to process I where CEST data were critical, the small chemical shift differences between states involved in processes III and IV (less than 1.5 ppm, see 'Materials and methods' 'Separating probes of processes I–IV') precluded analysis via CEST since major and minor state dips are, in general, poorly resolved. In these cases, however, high quality two-state fits of the CPMG dispersion data are obtained (*Figures 5A, 6A*).

In addition to the pair of complementary methodologies used in this study, a large number of different experiments are further necessary to separate the exchange events (*Table 2*). A detailed description of the approaches that have been used is given in 'Materials and methods'. Briefly, we have distinguished between intra- and inter-molecular processes by recording experiments as a function of protein concentration, separating process II which was concentration independent (electrostatic loop helix formation) from the other exchange events that showed increased minor state

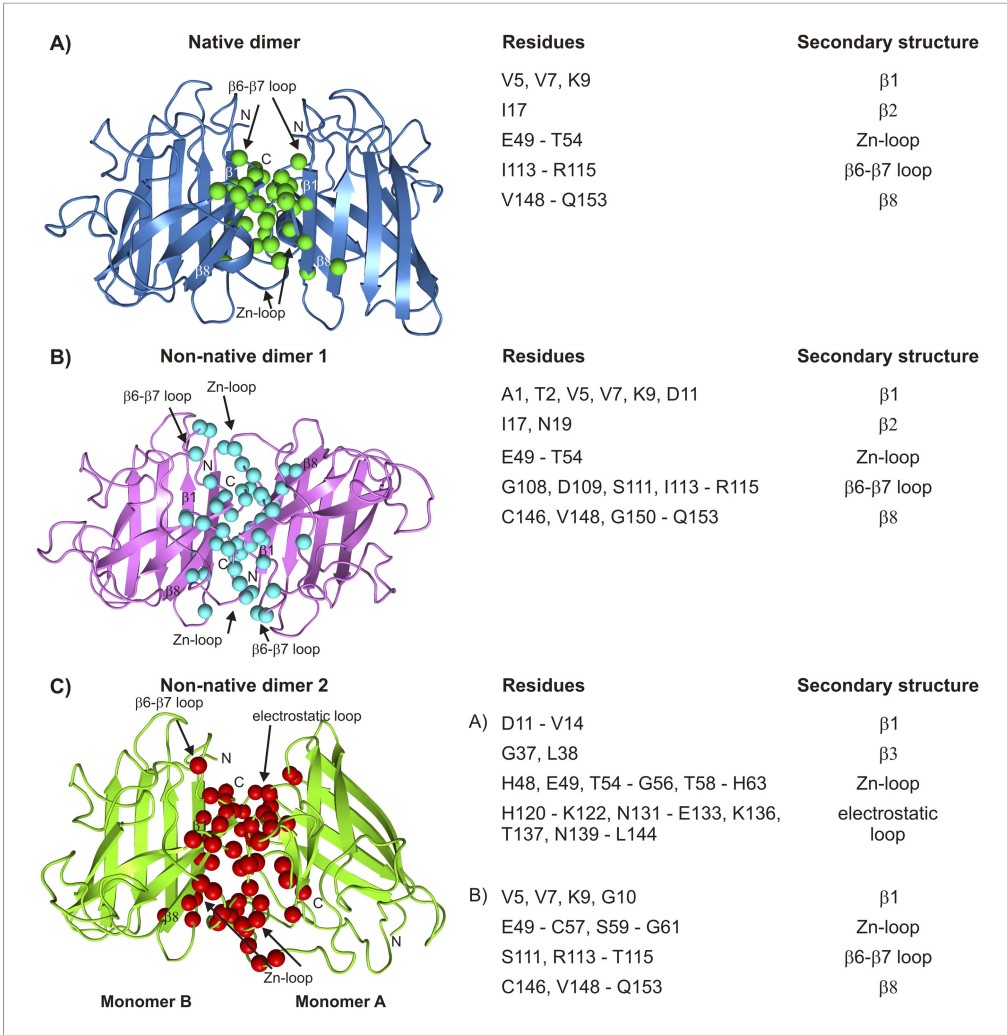

**Figure 10.** Interfaces of pWT apoSOD1$^{2SH}$ excited state dimers. The electrostatic and Zn-loops as well as strands β1 and β8 are key motifs in forming intermolecular interactions—both native and aberrant, as illustrated by the lowest energy structures for the native (**A**) and the two non-native dimers (**B**, **C**). Residues at the interface of the monomer units in each case were identified using a cutoff value of 2 Å$^2$ for the surface area buried upon dimerization and are shown as spheres and indicated in tabular form on the right. Note that the native dimer and the non-native dimer 1 are symmetric (axes of symmetry in the plane of the paper and perpendicular to the plane of the paper, respectively), while the non-native dimer 2 is not.

dips (CEST) or larger dispersion profiles (CPMG) at the higher protein concentrations. Temperature dependent studies were also performed that clearly distinguished processes I and III (increased CPMG dispersion profiles with higher temperature) from process IV (decreased profiles). Residues involved in processes I and III could be separated because different exchange parameters gave rise to profiles that could not be fit simultaneously. Finally, studies on G85R and A4V apoSOD1$^{2SH}$ have provided confidence in our assignment of residues to particular exchange events, since reporter residues of a particular process in the WT protein that is subsequently abolished via mutation no longer show evidence of conformational exchange. By means of example, process II is eliminated in G85R apoSOD1$^{2SH}$, while processes I, III, and IV can no longer be detected in A4V apoSOD1$^{2SH}$.

The combination of CEST and CPMG experiments described above leads to a kinetic scheme (*Figure 9B*), in which native apoSOD1$^{2SH}$ appears as a central hub for exchange processes leading to maturation (processes I, II) or aberrant association (processes III, IV). Experiments on G85R and A4V apoSOD1$^{2SH}$ have further clarified details of this kinetic network by providing insights into the

interconversion between different excited state conformations. For example, evidence that helix formation and dimerization are largely uncoupled is provided by mutations in apoSOD1$^{2SH}$ that eliminate either helix folding (G85R) or dimerization (A4V). The A4V mutant has no effect on helix folding, while G85R apoSOD1$^{2SH}$ does not influence dimerization processes I and III with only some effect on IV (*Figure 3A* and *Figure 3—figure supplement 1*). Therefore, helix formation is not on-pathway for formation of the native dimer or non-native dimer I from apoSOD1$^{2SH}$. In addition dimerization is not a prerequisite for helix formation (process II) since this would entail a concentration dependence for the helix process which is not observed (*Figure 4A*). Moreover, relaxation dispersion data from residues reporting on processes III and IV are all well fit to a two-state model. While this certainly does not prove that the exchange mechanisms for aberrant dimerization cannot be more complex, our present data do not justify such an interpretation. Accordingly, *Figure 9B* depicts the simplest kinetic scheme consistent with the spin relaxation data recorded on apoSOD1$^{2SH}$ where each of the four processes studied in this work has been shown separately. We cannot rule out the existence of a more complex network in which for instance the helical population dimerizes, the electrostatic loop in the dimeric population folds, or native and non-native dimers form larger assemblies. However, these processes involve populations of conformers that are below the detection limits of CEST and CPMG methods.

Having established the kinetic network and assigned residues to particular processes, structural models of the excited states were calculated using the molecular docking program HADDOCK (*Figures 9, 10*, see 'Materials and methods'). We have also attempted to supplement chemical shift restraint data with distances generated from paramagnetic relaxation enhancement (PRE) experiments by carrying out experiments involving mixtures of $^{15}$N-apoSOD1$^{2SH}$ and spin-labeled $^{14}$N-apoSOD1$^{2SH}$, as first described in other studies of SOD1 oligomerization (*Teilum et al., 2009*). However, these experiments were not successful because the exchange regimes for processes I, III, and IV ($k_{ex}$ = 150–600 s$^{-1}$) preclude the determination of PREs in the excited dimeric state, which can be reliably measured only for much faster processes ($k_{ex}$ > 10,000 s$^{-1}$) (*Iwahara and Clore, 2006*; *Clore, 2011*).

*Figure 10A–C* highlights the residues and secondary structure elements present at the interfaces of the native and non-native dimers that have been determined in this study. Despite the fairly low resolution of the calculated models important insights can, nevertheless, be obtained. For example, identification of the interfaces of the aberrant oligomers helps explain why Cu$_2$Zn$_2$SOD1$^{S-S}$ is so resistant to oligomerization, and potentially, why many of the ALS mutants localize to the dimer or the metal-binding region. The key elements of these non-native inter-molecular interfaces are strands β1, β8 and part of the Zn-binding loop, which are sequestered and hence protected upon formation of the native dimer, as well as the electrostatic loop, which is rigid in Cu$_2$Zn$_2$SOD1$^{S-S}$ (*Figure 2A*) and thus less likely to form promiscuous interactions. The protective nature of loop ordering is made clear in studies of S134N SOD1, which has a disordered electrostatic loop even in the holo state and has been suggested to form oligomers in solution (*Banci et al., 2005*). Crystals of this variant show non-native symmetric interactions whereby the disordered electrostatic loop from one monomer interacts with the cleft between strands β5–β6 of the other (*Elam et al., 2003*). Loss of metal and disruption of quaternary structure thus promote improper association both by destabilizing the protein and making higher energy conformations thermally accessible and by exposing regions of the protein most prone to forming these intermolecular interactions. It is noteworthy that the stabilizing disulfide bond in mutants of SOD1 is more easily reduced than in wild-type (*Tiwari and Hayward, 2003*), thus leading to larger concentrations of apoSOD1$^{2SH}$ and a higher tendency for forming aberrant oligomers.

Regions of apoSOD1$^{2SH}$ with millisecond timescale fluctuations that lead to the formation of non-native oligomers (processes III and IV) have been predicted to be aggregation-prone and to be involved in the initial stages of aggregation. For example, molecular dynamics (MD) simulations identified the N- and C-termini of SOD1, as well as residues I35-F45 and H110–H120, to have a pronounced tendency to aggregate (*Khare et al., 2005*). CPMG measurements recorded here show that many residues from the termini of SOD1 as well as from H110–H120 participate in both exchange processes III and IV. Moreover, nanosecond timescale simulations starting from dimeric SOD1 in the absence of bound metal show that the Zn and electrostatic loops become flexible and move away from one another, exposing strands β4 and β5. These observations led to the hypothesis that the initial events in SOD1 oligomerization may be triggered by the misfolded form of the protein possessing flexible Zn and electrostatic loops as well as exposed β4 and β5 strands (*Strange et al., 2007*). This is consistent with results from our experiments showing that conformational fluctuations involving

residues 45–46 (β4) and 85–87 (β5) as well as the dimer interface and parts of the electrostatic loop lead to non-native oligomerization (process IV).

Interestingly, the structural features of the non-native oligomers identified here are in keeping with expectations based on studies of soluble and insoluble SOD1 species in protein inclusions isolated from human patients (*Rakhit et al., 2007*; *Bosco et al., 2010*; *Forsberg et al., 2010*) and mouse models of ALS (*Rakhit et al., 2007*; *Bosco et al., 2010*; *Zetterström et al., 2013*). For example, SEDI antibodies raised against native–dimer interface residues 143–151, and for which there is no cross-reactivity with monomeric SOD1, could identify misfolded SOD1 in degenerating neurons of presymptomatic mice models of ALS, and also in mice overexpressing wild-type SOD1 (*Rakhit et al., 2007*). Notably, this region is exposed in non-native dimer II. In a second example, inclusions from neurons of sporadic ALS patients were immunoreactive to the C4F6 antibody raised against G93A SOD1, indicating that misfolded WT SOD1 species were present, since properly folded WT SOD1 is not reactive (*Bosco et al., 2010*). The C4F6 antibody recognizes residues 80–118 which are sequestered in the WT protein (*Strange et al., 2003*) but large stretches of which are exposed in the non-native oligomers that have been characterized here. Moreover, the importance of small dimeric particles in ALS disease pathology is made clear by a report in which 32 kDa SOD1 species were identified by biotinylation based chemical crosslinking of inclusion bodies obtained from spinal cords of sporadic and familial ALS patients (*Gruzman et al., 2007*).

Structural models for the non-native conformers described in this work provide further clues as to the molecular architecture of misfolded oligomers detected in vivo. Aggregates of SOD1 have been found to contain various protease resistant cores including regions 1–30, 90–120, and 135–153 (*Furukawa et al., 2010*), all of which contain residues present at the interfaces of the non-native oligomers identified presently, suggesting that these oligomers may serve as nuclei for downstream aggregation. Indeed, it is possible to dock asymmetric dimers formed via process IV to other subunits via native or non-native interfaces described here to generate higher order oligomers (*Figure 9—figure supplement 2*).

The presence of oligomeric conformations separated from native apoSOD1$^{2SH}$ by small free energy barriers under physiologically relevant conditions provides a compelling example of how improper protein assembly and aggregation can originate from locally unfolded or misfolded protein states accessible from the native free energy basin via thermal fluctuations. The oligomers reported here are formed at rates of ~100 s$^{-1}$, five orders of magnitude faster than the rate of unfolding (*Lindberg et al., 2004*), and are more likely to be sampled under physiological conditions in vivo than the globally unfolded conformation. Locally unfolded states have been implicated as key precursors in the aggregation of proteins (*Chiti and Dobson, 2008*) such as transthyretin (TTR) (*Lai et al., 1996*; *Quintas et al., 2001*; *Hammarström et al., 2003*), human lysozyme (*Canet et al., 2002*) and β2-microglobulin (*Jahn et al., 2006*) involved in other protein conformational disorders. The disruption of a stable quaternary TTR tetramer via mutations leads to a monomeric form which is aggregation-prone, and the subsequent formation of amyloid fibrils is associated with diseases such as senile systemic amyloidosis. In the case of TTR the less stable monomer transiently samples a non-native and potentially amyloidogenic conformation (*Lim et al., 2013*). This situation is not unlike that for SOD1 where demetallation and disulfide reduction destabilize the native dimer, resulting in a rugged landscape with thermally accessibly native and non-native conformations. It is noteworthy that the aberrant SOD1 oligomers identified here form without significant intra-molecular conformational rearrangements with most of the protein retaining its native structure, as is clear from the relatively small $^{15}$N chemical shift differences between ground and excited states associated with processes III and IV. Thus native apoSOD1$^{2SH}$, which is populated in the cell during the SOD1 maturation pathway (*Banci et al., 2012*), already appears to be poised to form aberrant interactions without the need for barrier crossing to a more pathological conformational state.

## Conclusions

Using NMR methods exquisitely sensitive to the presence of excited protein conformations (*Sekhar and Kay, 2013*), we have demonstrated that under physiologically relevant conditions immature monomeric apoSOD1$^{2SH}$ has a rugged free energy landscape on which mature conformations resembling the native state of metallated SOD1 coexist with aberrant non-native oligomers. The high propensity of the exposed dimer interface and the flexible electrostatic loop to self-associate results in the formation of three distinct oligomeric forms which exist in equilibrium with the native state of

apoSOD1$^{2SH}$. Structural models for the two non-native oligomers reported here may provide insight for the rational design of therapeutic agents to prevent the early stages of nascent SOD1 aggregation and ultimately formation of the cytotoxic species that leads to disease.

## Materials and methods

### Expression and purification of isotope-labeled Cu$_2$Zn$_2$SOD1$^{S-S}$ and apoSOD1$^{2SH}$

Genes encoding pWT SOD1 (*Getzoff et al., 1992*; *Vassall et al., 2011*; *Broom et al., 2014*) (pseudo wild-type protein where nonconserved Cys 6 and Cys 111 are replaced by Ala and Ser, respectively) or ALS-associated mutants G85R and A4V (*Valentine et al., 2005*) were inserted into the plasmid vector pHSOD1ASlacI$^q$ (*Getzoff et al., 1992*), with protein expression in BL21 *Escherichia coli* cells. Suitably labeled proteins were grown to 0.6–0.7 OD$_{600}$ in M9 minimal media (Na$_2$HPO$_4$ [6 g/l], KH$_2$PO$_4$ [3 g/l], NaCl [0.5 g/l], $^{15}$NH$_4$Cl [0.5 g/l], MgSO$_4$ [2 mM], $^{12}$C glucose [4 g/l for $^{15}$N-labeling], $^{13}$C glucose [2 g/l for U-$^{13}$C labeling] or 1-$^{13}$C glucose [2 g/l for selective labeling of methyl groups (*Lundström et al., 2007a*)], thiamine [0.0005 g/l], CaCl$_2$ [0.1 mM] with 100 mg/l ampicillin). Because Ile δ1 methyl groups are not efficiently $^{13}$C labeled when 1-$^{13}$C glucose is used as the $^{13}$C source, methyl $^{13}$C, 3,3-D$_2$α-ketobutyrate (50 mg/l) was also added (*Gardner and Kay, 1997*) approximately 1 hr prior to induction of protein overexpression. Protein expression was induced upon addition of 0.25 mM IPTG and the growth media were supplemented with CuSO$_4$ and ZnSO$_4$ (0.1 mM and 0.05 mM final concentrations, respectively) at this time. Expression continued for ~42 hr at 25°C after which cells were harvested by centrifugation. Purification of Cu$_2$Zn$_2$SOD1$^{S-S}$ was carried out using a modification of the procedure outlined in Getzoff et al. in which the DEAE column step was replaced by one involving a POROSHP2 column (*Getzoff et al., 1992*). Subsequent generation of apo-protein and disulfide reduction were achieved following protocols described previously (*Vassall et al., 2006, 2011*). The metal content in samples was determined by Inductively Coupled Plasma Atomic Emission Spectroscopy. Reduction of the disulfide linking Cys 57 to Cys 146 (apoSOD1$^{2SH}$) was verified by making use of the large differences in $^{13}$Cβ chemical shift values of Cys residues in oxidized (on average 40.7 ± 3.8 ppm) and reduced (28.4 ± 2.4 ppm) states (*Sharma and Rajarathnam, 2000*). The measured values for Cys 57 and Cys 146 of 28.1 ppm and 32.5 ppm, respectively, are consistent with reduced residues.

### NMR sample preparation

Samples of isotope-labeled apoSOD1$^{2SH}$ were prepared in buffer containing 20 mM HEPES pH 7.4, 1 mM TCEP, 1 mM NaN$_3$, and 90% H$_2$O/10% D$_2$O (NMR buffer), while a 0.8 mM sample of Cu$_2$Zn$_2$SOD1$^{S-S}$ was prepared in buffer containing 20 mM HEPES pH 7.4, 5 mM isoascorbate and 90% H$_2$O/10% D$_2$O. $^{15}$N-labeled samples of apoSOD1$^{2SH}$ ranged in concentration from 0.7 mM to 3.5 mM, the concentration of U-[$^{13}$C,$^{15}$N] apoSOD1$^{2SH}$ was 1.5 mM and the selectively methyl-$^{13}$C labeled samples were prepared at concentrations of 2.5 mM and 0.8 mM. Suprasil NMR tubes (Wilmad Lab-glass, cat. no. 535-PP-7SUP) were used in recording all spectra for apoSOD1$^{2SH}$ samples unless stated otherwise. After placing the apoSOD1$^{2SH}$ sample inside the NMR tube, the tube was immediately purged with Ar and sealed before use.

### NMR experiments and data processing

NMR spectra were acquired using 11.7 T ($^1$H frequency of 500 MHz), 14.0 T (600 MHz) or 18.8 T (800 MHz) Varian INOVA spectrometers, with the 600 MHz spectrometer equipped with a cryogenically cooled probe. All NMR experiments were carried out at 25°C unless stated otherwise, with the temperature in the sample chamber measured using a thermocouple attached to a digital thermometer placed inside an NMR tube containing D$_2$O. NMR data sets were processed with the NMRPipe (*Delaglio et al., 1995*) suite of programs and visualized using Sparky (*Goddard and Kneller, 2006*). Peak intensities were extracted using either NMRPipe or by fitting the respective peak lineshapes using FuDA (http://pound.med.utoronto.ca/software).

### Backbone and sidechain resonance assignment

Experiments for resonance assignments were recorded at 600 MHz using a U-[$^{13}$C,$^{15}$N] apoSOD1$^{2SH}$ sample. Assignments of backbone and sidechain resonances were obtained using a combination of 2D $^1$H-$^{15}$N HSQC and $^1$H-$^{13}$C HSQC data sets as well as 3D HNCACB, CBCA(CO)NH, HNCO,

HN(CA)CO, (H)CCA(CO)NH, and H(CCACO)NH experiments, described in detail previously (*Cavanagh et al., 1995*; *Sattler et al., 1999*). Assigned chemical shifts of pWT apoSOD1$^{2SH}$ have been deposited to the Biological Magnetic Resonance Data Bank (BMRB accession number 26570). Assignments of $^1H^N$ and $^{15}N$ resonances for G85R and A4V mutants were transferred from pWT using $^{15}N$ NOESY-HSQC (*Zhang et al., 1994*) data sets.

## Translational diffusion experiments

Translational diffusion coefficients of apoSOD1$^{2SH}$ and Cu$_2$Zn$_2$SOD1$^{S–S}$ were determined using a 1D pulsed field gradient $^{15}N$-edited longitudinal encode-decode (LED) scheme (*Choy et al., 2002*). The total diffusion delay used in the measurements was 150 ms, and encoding and decoding gradient durations were 1 ms each. Each dataset comprised 15 gradient strengths ranging between 4–60 G/cm with three duplicate points for error analysis. The amide region (7.8–9.4 ppm) of each 1D $^{15}N$-edited spectrum was integrated as a single unit and the variation in the resulting intensity (I) with squared gradient strength (G$^2$) fit to a single exponential decay of the form, I = I0 exp(–dG$^2$), to extract a diffusion decay rate (d) which is linearly proportional to the translational diffusion coefficient of the protein.

## $^{15}N$ R$_1$, R$_{1\rho}$, and $^1H$-$^{15}N$ heteronuclear NOE

In order to estimate the rotational correlation times of apoSOD1$^{2SH}$ and Cu$_2$Zn$_2$SOD1$^{S–S}$, $^{15}N$ spin relaxation rates, R$_1$ and R$_{1\rho}$, and steady state $^1H$-$^{15}N$ heteronuclear NOE values were measured using pulse sequences described earlier (*Farrow et al., 1994*; *Korzhnev et al., 2002*; *Ferrage et al., 2010*). 11 relaxation delays ranging between 10–600 ms and 10 delays between 3–80 ms were used for the R$_1$ and R$_{1\rho}$ experiments, respectively. Three relaxation delays were repeated in each of the experiments for error analysis. Residue-specific R$_1$ and R$_{1\rho}$ values were obtained from fits of peak intensities vs relaxation time to a single exponential decay function, while NOE ratios were ascertained directly from intensities in experiments recorded with (5 s relaxation delay followed by 7 s saturation) and without (relaxation delay of 12 s) saturation. Errors in NOE values were calculated by propagating the error in the respective peak intensities. R$_{1\rho}$ values were converted to R$_2$ rates using the relation, R$_{1\rho}$ = R$_2$cos$^2\theta$ + R$_1$sin$^2\theta$, where $\theta$ = tan$^{-1}$ ($\delta$/B$_{SL}$), B$_{SL}$ is the spin lock field strength (Hz), $\delta$ is the resonance offset of the spin in question (Hz), and $\theta$ is the angle made by the effective field with the direction of B$_{SL}$ irradiation.

Values of rotational correlation times were obtained from fits of (R$_1$, R$_2$, NOE) to standard expressions (*Abragam, 1961*; *Kay et al., 1989*) with spectral density functions as defined by the model free approach of Lipari and Szabo (*Lipari and Szabo, 1982*). Only $^{15}N$ spins with R$_{ex}$ < 2 s$^{-1}$, as measured in $^{15}N$ CPMG experiments recorded at 600 MHz (25°C, 3.5 mM protein concentration), were used in the analysis. Residues were included only if they are in ordered regions of the protein and additionally for apoSOD1$^{2SH}$ if chemical shift-derived order parameters (*Figure 2A*) were larger than ~0.6. All fits used the software package Modelfree 4.15 (*Mandel et al., 1995*) and for simplicity isotropic diffusion tensors were assumed with residue specific parameters that include the order parameter squared, S$^2$, and a timescale describing fast motion, $\tau_e$. Overall, 15 (localized to 5 of the 8 strands) and 112 residues were considered in the analyses for apoSOD1$^{2SH}$ and Cu$_2$Zn$_2$SOD1$^{S–S}$, respectively, and correlation times of 11.6 ns (apoSOD1$^{2SH}$) and 18.2 ns (Cu$_2$Zn$_2$SOD1$^{S–S}$) were obtained. Note that the small number of apoSOD1$^{2SH}$ residues selected for analysis reflects the widespread chemical exchange in this variant.

## CEST

All CEST experiments were recorded using previously published pulse schemes (*Vallurupalli et al., 2012*; *Vallurupalli and Kay, 2013*; *Long et al., 2014*) on samples labeled with $^{15}N$ ($^{15}N$ CEST) or U-[$^{15}N$,$^{13}C$] ($^{13}C'$, $^{13}C\alpha$ and $^{13}CH_3$ CEST). Relaxation delays (T$_{CEST}$) of 350 ms ($^{15}N$), 300 ms ($^{13}C'$), 125 ms ($^{13}C\alpha$), and 250 ms ($^{13}CH_3$) were employed and CEST profiles were recorded using 1 or 2 B$_1$ radiofrequency fields for $^{15}N$ (31 and 59 Hz), 1 B$_1$ field for $^{13}C\alpha$ (27 Hz) and 2 B$_1$ fields for both $^{13}C'$ (26 and 42 Hz) and $^{13}C$-methyl (15 and 24 Hz) spins. B$_1$ field strengths were calibrated as reported earlier (*Vallurupalli et al., 2012*; *Vallurupalli and Kay, 2013*). A series of 2D spectra were recorded with the B$_1$ field stepped from 93.9–136.9 ppm for $^{15}N$ (89 planes at B$_1$ = 31 Hz and 54 planes at 59 Hz), 169–180.9 ppm for $^{13}C'$ (82 planes at 26 Hz and 50 planes at 42 Hz), 41.2–68.6 ppm for $^{13}C\alpha$ (120 planes) and 9.3–30 ppm for $^{13}CH_3$ (139 planes at 15 Hz and 107 planes at 24 Hz). CEST profiles were generated as

the ratio in intensities of peaks in spectra acquired with and without the $T_{CEST}$ period vs the position of the low power $B_1$ field (*Fawzi et al., 2011*; *Vallurupalli et al., 2012*). Errors were determined from the scatter in the baseline of CEST profiles (*Bouvignies et al., 2014*). Chemical shifts of the excited state were obtained directly from the CEST curves by fitting them to a sum of Gaussians using the 'curvefit' routine in Matlab (The Mathworks Inc., Natick, MA). In order to extract exchange parameters, CEST profiles were fit to two- or three-state exchange models using the home written program Chemex (https://github.com/gbouvignies/chemex). Errors in fitted values were extracted using the covariance matrix method (*Taylor, 1997*).

## CPMG relaxation dispersion

$^{15}$N CPMG data sets were recorded on U-$^{15}$N, U-[$^{15}$N,$^{13}$C] and selectively $^{13}$C-labeled samples produced with 1-$^{13}$C glucose, while $^{13}$C methyl profiles were measured on selectively $^{13}$C-labeled samples only. CPMG experiments were generally, but not always, recorded at two static magnetic field strengths. Constant-time CPMG relaxation elements were used for both $^{15}$N (*Vallurupalli et al., 2007*) and $^{13}$CH$_3$ (*Lundström et al., 2007b*) experiments with relaxation delays ($T_{CPMG}$) of 30 ms, approximately 20 different values of the CPMG pulsing frequency ranging from 33.3–1000 Hz for $^{15}$N, from 66.6–2000 Hz for $^{13}$CH$_3$ and 3–5 duplicate points acquired for error analysis (*Korzhnev et al., 2007*). Peak intensities in CPMG experiments were converted to effective transverse relaxation rates ($R_{2,eff}$) using the equation, $R_{2,eff} = (-1/T_{CPMG}) \ln(I/I_0)$, where I and $I_0$ are peak intensities measured with and without the CPMG delay (*Korzhnev et al., 2007*). Errors in intensities determined from duplicate points were propagated to obtain errors in $R_{2,eff}$ (*Korzhnev et al., 2007*). The variation in $R_{2,eff}$ as a function of CPMG pulsing frequency (so called relaxation dispersion profile) was fit to a two-state model of exchange using in-house-written software CATIA (http://pound.med.utoronto.ca/software), while Chemex was used for global fits of CEST and CPMG data to both two- and three-state exchange models. Both programs numerically propagate the Bloch–McConnell equations (*McConnell, 1958*) relevant for a particular model of exchange. Errors in fitted values of populations, rate constants and chemical shift changes were determined using the covariance matrix method (*Taylor, 1997*).

## Metal independence of processes I and II

Transient dimerization (process I) transforms apoSOD1$^{2SH}$ into a conformation which is very similar to the native ground state of metal-bound Cu$_2$Zn$_2$SOD1$^{S-S}$ while the local folding of the electrostatic loop, residues 130–140 (process II), generates a helix which is found in the native dimeric Cu$_2$Zn$_2$SOD1$^{S-S}$ state. Because of the high affinity of SOD1 for Cu and Zn, (dissociation constants smaller than ~$10^{-13}$ and ~$10^{-17}$ M, respectively [*Crow et al., 1997*]) it is necessary to establish that processes I and II are not 'simply' the result of apoSOD1$^{2SH}$ binding to trace metals present in the sample. As illustrated in *Figure 3—figure supplement 3*, a number of controls rule out that processes I and II are metal-dependent exchange events, reflecting rather the landscape of metal free apoSOD1$^{2SH}$ and the propensity of this state to adopt 'native-like' conformations. We summarize the evidence that processes I and II are metal independent in the points below.

1. After recording spectra of samples of pWT apoSOD1$^{2SH}$ (Suprasil tubes) metal analysis was carried out via Inductively Coupled Plasma Atomic Emission Spectroscopy (ICP-AES). No detectable levels of Cu or Zn could be found (detection limits of 750 nM for Zn and 300 nM for Cu; *Figure 3—figure supplement 3A*). Among those elements tested (Ag, Al, As, B, Ba, Be, Ca, Cd, Co, Cr, Cu, Fe, K, Mg, Mn, Mo, Na, Ni, Pb, Sb, Se, Si, Ti, Tl, V, and Zn) only B (104 µM), Ca (2.25 µM), K (24.6 µM), and Na (17.3 mM) could be detected, with Ca the only divalent metal. Its concentration is an order of magnitude smaller than what would be required to give rise to the excited state populations observed (2–3%), since for the protein concentration of 1.3 mM used in the analysis approximately 30 µM of metal would be necessary for the excited state to correspond to a Ca-bound state. This is further supported by the fact that very similar CEST profiles were obtained from samples that had been placed in regular glass NMR tubes where the Ca concentration was determined to be below the detection limit of ICP-AES (*Figure 3—figure supplement 3B,C*).
2. To further confirm that CEST profiles observed for residues reporting on processes I and II are not the result of direct Zn binding we added ~140 µM Zn to a Zn and Cu-free apoSOD1$^{2SH}$ sample in a Suprasil tube. For many residues, two sets of peaks are observed in $^1$H-$^{15}$N HSQC spectra of the resultant sample due to the slow exchange between apoSOD1$^{2SH}$ and the metal-bound form of the protein (*Figure 3—figure supplement 3D*; representative peaks indicated with *). The CEST profiles from apoSOD1$^{2SH}$ are, however, very similar to those recorded in the absence of metal

(*Figure 3—figure supplement 3E,F*). If the CEST profiles quantified were reporting on Zn binding the CEST dips would increase significantly to reflect an excited state population of ~20% (140 μM Zn, 700 μM protein). The dips, however, do not change in size, confirming that the excited state is independent of Zn binding. Our results are consistent with expectations based on previous studies where the half-life for Zn release has been determined to be at least 11 hr (*Crow et al., 1997*), orders of magnitude longer than is required for observation of chemical exchange via CEST where exchange rates of at least 50 s$^{-1}$ are needed (*Vallurupalli and Kay, 2013*).

## Concentration and temperature-dependence of R$_{ex}$

Residue- and nucleus-specific R$_{ex}$ values obtained from $^{15}$N and $^{13}$C-methyl CPMG experiments were calculated as the difference between R$_{2,eff}$ rates at the lowest and highest CPMG pulsing frequencies (*Palmer et al., 2000*). In order to evaluate the protein concentration-dependence of R$_{ex}$, data recorded at either 3.5 mM ($^{15}$N) or 2.5 mM ($^{13}$CH$_3$), 800 MHz, were analyzed and residues with R$_{ex}$ > 5 s$^{-1}$ selected. For this subset, ratios (R) of R$_{ex}$ values measured at protein concentrations of 3.5 mM and 0.7 mM ($^{15}$N) or 2.5 mM and 0.8 mM ($^{13}$CH$_3$), 600 MHz, were calculated and the error in each R value, $\sigma_R$, obtained by propagating errors in R$_{ex}$ at the two concentrations. Residues for which R > 1.4 and R/$\sigma_R$ ≥ 3 were classified as part of the concentration dependent group. The temperature-dependencies of CPMG profiles for residues with concentration-dependent R$_{ex}$ values were determined by examining R$_{ex}$ rates measured in either 2 mM ($^{15}$N) or 2.5 mM ($^{13}$CH$_3$) samples at 25°C and 35°C, 800 MHz. Residues with temperature-dependent R$_{ex}$ values were defined as those with D = |R$_{ex}$(35°C) − R$_{ex}$(25°C)| > 1 s$^{-1}$ and D/$\sigma_D$ ≥ 3, where $\sigma_D$ is the error in D obtained by propagating errors in R$_{ex}$ at 25° and 35°C. A total of 42 residues were selected from analysis of the protein concentration dependencies via these criteria and these were sorted into processes III (23 residues) and IV (19 residues) based on the temperature dependence criteria described above.

## Three-state fitting of CEST and CPMG data

CPMG and CEST data sets recorded on pWT apoSOD1$^{2SH}$ and reporting on processes I and II could not be fit simultaneously to a two-state model of chemical exchange. (In contrast, dispersion data from process III and IV were well fit to a two state model.) As a result a three-state model was chosen in the analysis of processes I and II, with fits performed in several stages. First, a number of residues whose $^{15}$N CEST and CPMG profiles indicated large chemical shift differences between the ground and excited states were identified for each process (Gly 51, Asn 53, Thr 54 and Gly 61 for process I and Thr 135 and Thr 137 for process II); these residues were used in subsequent analyses to extract exchange parameters. Next, a complete set of three-state models was chosen making use of the fact that the chemical shifts for probes in two of the three states are known from CEST profiles, referred to as the ground (N) and the excited state (H) in what follows. Note that only two dips are observed in the CEST profiles, suggesting that the third state has similar chemical shifts to either N or H. There are, thus, three classes of linear three-state models that can be selected, depending on the relative position of the hidden intermediate (I; not observed in the CEST profile in the 'reaction scheme'), as illustrated below. These include, $N \underset{k_{IN}}{\overset{k_{NI}}{\rightleftharpoons}} I \underset{k_{HI}}{\overset{k_{IH}}{\rightleftharpoons}} H$ [1], $I \underset{k_{NI}}{\overset{k_{IN}}{\rightleftharpoons}} N \underset{k_{HN}}{\overset{k_{NH}}{\rightleftharpoons}} H$ [2], $N \underset{k_{HN}}{\overset{k_{NH}}{\rightleftharpoons}} H \underset{k_{IH}}{\overset{k_{HI}}{\rightleftharpoons}} I$ [3], where I is on-pathway [1] or off-pathway and originating either from the ground [2] or the excited states [3]. In each of these three classes, the $^{15}$N chemical shift of I can be close to N or to H (hence explaining the appearance of only two dips in each CEST trace), so that a total of six initial models was considered. The CEST and CPMG data were not of sufficiently high quality to distinguish the triangular model with all three states connected from the linear schemes described above, so the simplest linear 3-state schemes were selected. For each model, $A \underset{k_{BA}}{\overset{k_{AB}}{\rightleftharpoons}} B \underset{k_{CB}}{\overset{k_{BC}}{\rightleftharpoons}} C$, the search in parameter space was carried out across 2 rate constants, $k_{AB,ex} = k_{AB} + k_{BA}$ and $k_{BC,ex} = k_{BC} + k_{CB}$, and two populations (corresponding to the sparsely populated states H and I, $p_H$ and $p_I$ in schemes [1]–[3] above). For transient dimerization (process I), the grid consisted of 11 points in $k_{AB,ex}$ and $k_{BC,ex}$ logarithmically spaced to span the range 10–2016 s$^{-1}$, while the grid in $p_H$ and $p_I$ comprised 16 points linearly spaced between 0.5 and 23%. For the local folding process II, the grid in $p_H \times p_I \times k_{AB,ex} \times k_{BC,ex}$ consisted of $10 \times 10 \times 16 \times 16$ points between 0.5 and 14% in $p_H$ and $p_I$ (linear grid) and 10 and 11,529 s$^{-1}$ in rates (logarithmic grid). At each grid point $\chi^2_{red}$ was calculated by fitting CEST and CPMG profiles on a residue-specific basis using Chemex. Values of $k_{AB,ex}$, $k_{BC,ex}$, $p_H$ and $p_I$ along with $\Delta\varpi_{NH}$ (that can be obtained directly from the positions of the major and minor dips in the CEST profiles) were kept fixed in each fit, while $\Delta\varpi_{NI}$ was allowed to vary along with the relaxation rates R$_{2N}$, R$_{2H}$, R$_{2I}$, and R$_{1N}$.

Here, $\Delta\varpi_{NH}$ and $\Delta\varpi_{NI}$ are the chemical shift differences for the corresponding residues in the ground state (N) and either the excited (H) or hidden intermediate (I) states, respectively, while $R_{2N}$, $R_{2H}$, and $R_{2I}$ are transverse relaxation rates of the ground, excited and hidden states respectively and $R_{1N}$ is the longitudinal relaxation rate of N. For each three state-model and for each residue the grid search protocol described above generated a $\chi^2_{red}$ surface as a function of $k_{AB,ex}$, $k_{BC,ex}$, $p_H$, and $p_I$. The surfaces for all residues of a specific process were then added and the top 10 values of $k_{AB,ex}$, $k_{BC,ex}$, $p_H$, and $p_I$ (i.e., smallest $\chi^2_{red}$) for each of the six models were used as starting points in the next stage of $\chi^2_{red}$ minimizations. Here, all parameters were allowed to vary and the CEST and CPMG data for the residues indicated above associated with a particular process fit globally to extract final 'optimal' values of $k_{AB,ex}$, $k_{BC,ex}$, $p_H$, $p_I$, $\Delta\varpi_{NH}$, and $\Delta\varpi_{NI}$ and the best models for describing the data. As described in 'Separating probes of processes I–IV' below high quality fits of the combined CEST/CPMG data using this approach could be obtained for residues reporting on process II but not I.

## Separating probes of processes I–IV

### Process I: transient native-like dimerization

In initial $^{15}N$ CEST studies of apoSOD1$^{2SH}$ we noted that a large minor dip was present in profiles for G61 approximately 8 ppm upfield from the resonance position of the major conformer (*Figure 3A*). Previous studies of ground states of monomeric (*Banci et al., 2003*) and dimeric (CMD and EMM, unpublished) forms of apoSOD1$^{S–S}$ clearly show that the $^{15}N$ chemical shift of G61 moves from ~108 ppm to ~100 ppm upon dimerization. The presence of such a similarly large chemical shift change for G61 in an excited state of apoSOD1$^{2SH}$ provides strong evidence that the transition observed via CEST is a dimerization process to a native-like conformation. In addition to G61, three other residues were identified with minor state dips that were well separated from those of the ground state (G51: 8.3 ppm, N53: 8.1 ppm, T54: 3.8 ppm chemical shift changes), and all localized to the same region of the protein structure. As with G61, these chemical shift differences were essentially identical to those observed between monomeric and dimeric apoSOD1$^{S–S}$ in conventional NMR studies of ground state conformers. The $^{15}N$ CEST profiles for G51, N53, T54, and G61 (*Figure 3* and *Figure 3—figure supplement 1*) are all clearly protein concentration-dependent, consistent with a dimerization event. Because these residues localize to a region of the protein that comprises the dimer interface in Cu$_2$Zn$_2$SOD1$^{S–S}$ and report on native dimerization, we included additional residues as reporters so long as a decrease in solvent accessible surface area of greater than 2 Å$^2$ was calculated between the monomer and the native dimer. A complete list of residues selected in this manner is provided in *Table 3*. Notably, $R_{ex}$ values obtained from $^{15}N$ CPMG measurements for the four residues reporting on process I for which the largest dispersions were obtained (G51, N53, T54, and G61:maximum of 3 s$^{-1}$, see below) displayed the same temperature dependence (increase in $R_{ex}$ with increasing temperature), as would be expected for probes reporting on the same exchange event.

Interestingly, $^{15}N$ CEST profiles for residues involved in process I in pWT apoSOD1$^{2SH}$ (that included 2 B$_1$ fields—31 and 59 Hz) could not be well fit together using a two-state exchange model even at the single residue level (*Figure 3—figure supplement 2A*). Therefore, additional fits involving three-state models, as described above in 'Three-state fitting of CEST and CPMG data' were performed, based on an extensive grid search of parameter space to obtain optimum rate constants of exchange and populations of the interconverting states. However, combined analysis of $^{15}N$ CEST and $^{15}N$ CPMG profiles for residues G51, N53, T54, and G61 (largest $\Delta\varpi$ values) could not simultaneously be well fit to a three-state model. This is not unexpected given that the native dimer interface participates in three different oligomerization processes (I, III, and IV), as discussed in the text. In order to estimate the excited state dimer population and its lifetime for the pWT protein, residue-specific CEST data at each B$_1$ field were fitted separately (*Figure 3—figure supplement 2B*). In this case, fits improve significantly and estimates for the excited state population range from 1.5–4.5% (1.3 mM protein concentration) and its lifetime from 2.2–4.5 ms. By means of comparison, high quality two-state fits of $^{15}N$ CPMG data from residues reporting on process I in the G85R mutant (in which process II is eliminated and process IV decreases significantly) were obtained (*Figure 7—figure supplement 1A*) and the resultant $\Delta\varpi$ values correlate well with those obtained via CEST. Further, CPMG profiles acquired at 2 B$_o$ fields for residues N53 and T54 of pWT apoSOD1$^{2SH}$ reporting on process I could also be fit well (together) to a two-state model and $\Delta\varpi$ values derived from the fit are in excellent agreement with those obtained from CEST measurements.

Despite the fact that the exchange dynamics at the level of the dimer interface are complex, all CEST profiles showed only a single minor dip and its position (and hence the chemical shift of the probe in the excited state) was determined robustly, even when the exchange parameters were not. Excellent agreement is observed between excited state chemical shifts of residues selected as part of process I, from analysis of CEST data, and the previously reported (*Banci et al., 2002a*) ground state chemical shifts of dimer interface residues in Cu$_2$Zn$_2$SOD1$^{S-S}$ (*Figure 3*). This provides strong evidence that the selected residues collectively report on the same native dimerization process, although there is some 'contamination' from other processes that involve residues in the vicinity.

## Process II: transient native-like formation of the electrostatic loop helix

The backbone $^{15}$N chemical shifts of T135 and T137 are strikingly different between apoSOD1$^{2SH}$ and Cu$_2$Zn$_2$SOD1$^{S-S}$, providing a strong signature for the formation of the electrostatic loop helix that accompanies Zn binding (*Banci et al., 2002b*, *2003*). Large CEST dips corresponding to a minor state conformation were observed in pWT apoSOD1$^{2SH}$ profiles for both T135 and T137 that match the expected positions of a state in which the helix is present (~10 ppm upfield from resonance positions of the major conformation in both cases). This provides strong evidence for an excited state in which the electrostatic loop helix is formed. Amino acids localized to the electrostatic loop (residues 130–140) were thus selected as reporting on process II (*Table 3*). Focusing on $^{15}$N CEST profiles for residues in this region with large separations between major and minor dips ($\Delta\varpi$ values > 3 ppm) clearly establishes that the sizes of the minor dips are independent of protein concentration, as expected for a unimolecular process (*Figure 4* and *Figure 4—figure supplement 1*). In addition, all the minor state $^{15}$N CEST dips assigned to this process disappeared in the G85R mutant, as would be expected for residues reporting on the same event. Further, in A4V apoSOD1$^{2SH}$ minor dips in $^{15}$N CEST profiles involving residues reporting on process II could be distinctly observed, though those for the concentration-dependent processes I, III, and IV were no longer present. This provides further evidence that the residues selected all report on the same process.

As with data from process I, CEST and CPMG profiles probing transient helix formation in pWT apoSOD1$^{2SH}$ could not be well-fit together assuming a two-state model of chemical exchange, even on a per-residue basis (*Figure 3—figure supplement 2C*), although such data were well-fit to a two-state model in the case of the A4V mutant. However, in the case of pWT apoSOD1$^{2SH}$ it was possible to generate high quality three-state fits of CEST (2 B$_1$ fields) and CPMG profiles (2 B$_0$ fields) simultaneously for Thr 135 and Thr 137, two residues with very large $\Delta\varpi$ values (~10 ppm), using a procedure described above in 'Three-state fitting of CEST and CPMG data', with the results summarized in *Figure 3—figure supplement 2D,E*.

## Process III: transient non-native dimer I formation

Amide $^{15}$N or methyl $^{13}$C probes were selected as reporters of process III (*Table 3*) if: (i) R$_{ex}$ values calculated from $^{15}$N or $^{13}$C CPMG profiles all increased with increasing protein concentration, consistent with an oligomerization process (*Figure 5C*, *Figure 5—figure supplement 1A*); (ii) R$_{ex}$ values increased with increasing temperature (*Figure 5D*, *Figure 5—figure supplement 1C*); (iii) CPMG dispersion profiles of residues collectively could be well fit to a two-site model of chemical exchange.

The temperature dependence of R$_{ex}$ values distinguishes probes that report on processes III and IV (*Figures 5D, 6D*, *Figure 5—figure supplement 1C,D*). However, because R$_{ex}$ values increase with increasing temperature for residues associated with processes I and III we sought an additional criterion for separating these probes. For distinguishing between $^{15}$N probes of processes I and III, we focused on $^{15}$N CPMG experiments recorded on a 1.5 mM sample of G85R apoSOD1$^{2SH}$ and, in particular, on profiles from G51, N53, T54, and G61, residues that report on the native dimerization process (see 'Process I: Transient native-like dimerization', above) with large $\Delta\varpi$ values (approximately 8 ppm for G51, N53 and G61 and 4 ppm for T54). For these residues, the exchange is slow on the chemical shift timescale (see below) and dispersion profiles are well fit together to a model of two-site chemical exchange, $E \underset{k_{GE}}{\overset{k_{EG}}{\rightleftarrows}} G$, R$_{ex}$ is given to excellent approximation by the relation (*Palmer et al., 2000*),

$$R_{ex} = p_G p_E k_{ex} \frac{(\Delta\omega)^2}{k_{ex}^2 + (\Delta\omega)^2}, \tag{1}$$

where p$_E$ = (1 − p$_G$) is the fractional population of the excited state, k$_{ex}$ = k$_{EG}$ + k$_{GE}$ and $\Delta\omega$ is the difference in resonance frequencies (rad/s) of a spin in states E and G. *Equation 1* makes it clear that

$R_{ex}$ is a maximum when $k_{ex} \ll \Delta\omega$, that is in the slow exchange regime. For residues G51, N53, T54 and G61, a maximum $R_{ex}$ value of 2.8 s$^{-1}$ is calculated using *Equation 1* and the best fit exchange parameters and chemical shifts, that provides an upper bound on the magnitude of the dispersion profiles that can be expected for process I. Notably, calculated (and measured) values of $R_{ex}$ are field independent as expected in the slow exchange regime (*Palmer et al., 2000*). $R_{ex}$ values calculated for G85R apoSOD1$^{2SH}$ using *Equation 1* for all amides chosen as reporting on process III, satisfying criteria (i)–(iii) listed above, are greater than 4 s$^{-1}$. A plot of $R_{ex}$ values for those residues included in process III (*Table 3*) is provided in *Figure 7—figure supplement 1D* (mean value of 5.2 ± 0.3 s$^{-1}$), along with those for G51, N53, T54, and G61 from process I, clearly illustrating that processes I and III are distinct. The observed differences in $R_{ex}$ values for the processes are expected based on the fact that profiles from residues reporting on the two exchange events could not be fit well together (*Figure 7—figure supplement 1A,B*).

The approach described above allows a rigorous separation of $^{15}$N probes of processes I and III and we have used a similar procedure for methyl-$^{13}$C probes. Because methyl $^{13}$C CPMG data were recorded only on samples of pWT apoSOD1$^{2SH}$ an estimate of the maximum $R_{ex}$ value for spins reporting on process I was obtained for the pWT form (4.4 s$^{-1}$, 2.5 mM sample) using $^{15}$N dispersion profiles from N53 and T54 of pWT apoSOD1$^{2SH}$ that are well fit to a two-state model (and report on process I, see above). Methyl-$^{13}$C dispersions with $R_{ex}$ values larger than this limiting value (recorded using the same 2.5 mM sample) cannot derive from process I and, so long as they satisfy criteria (i)–(iii) above, were assigned as probes of process III.

Notably, chemical shift differences between exchanging states were smaller for process III (and IV, see below) than for processes I and II (typically $\Delta\varpi < 1.5$ ppm for III and IV) so that, in general, the minor state dips in CEST profiles were not well resolved from those of the major conformer. As a result relaxation dispersion data proved to be more valuable for the extraction of exchange parameters. Analysis of the resultant CPMG profiles (recorded at 600 and 800 MHz) was based on a dimerization model (*Palmer et al., 2000*), $2M \underset{k_{DM}}{\overset{k_{MD}}{\rightleftharpoons}} D$, to extract the population of the excited state, $p_D$ and rate of interconversion between M and D, $k_{ex}$. These in turn can be recast as $k_{ex} = 2 k_{MD} [M] + k_{DM}$ and $p_D = 2 [D]/(2 [D] + [M]) = 2 k_{MD} [M]/(2 k_{MD} [M] + k_{DM})$ to obtain $k_{MD}$, $k_{DM}$ and hence the equilibrium constant for dimer formation, $K_{MD}$, for a known total protein concentration (*Figure 7*). In all the analyses, data recorded at separate temperatures (20°C–35°C) were fitted independently, with chemical shift differences, $\Delta\varpi_{MD}$, also allowed to vary for each temperature.

## Process IV: transient non-native dimer II formation

Residues used as probes of process IV (*Table 3*) were selected following similar criteria to those listed for process III with the exception that $R_{ex}$ values showed a decrease with increasing temperature (*Figure 6D*, *Figure 5—figure supplement 1D*). Temperature-dependent populations ($p_D$) and rate constants ($k_{ex}$) for oligomerization (assumed dimerization, see below) were determined from global fits of $^{15}$N CPMG data at 600 and 800 MHz to a two-state model as described for process III. On the basis of structural models determined by HADDOCK (*Dominguez et al., 2003*) (see *Figure 9—figure supplement 1*) the dimer was found to be asymmetric.

In general, asymmetric dimerization is described in terms of a three-state exchange process, whereby a particular nucleus can have different chemical shifts in the monomer ($\omega_M$) and in each of the monomers that comprise the dimer ($\omega_{D1}$, $\omega_{D2}$). However, for the particular case where the environment of the spin probe changes in only one of the monomers of the dimer, as in the case of process IV, it is reasonable to assume that $\omega_{D2} = \omega_M$. In this case, and if the differences in intrinsic transverse relaxation rates of the monomer and dimer are not significantly different, the three-state Bloch–McConnell equations for transverse magnetization ($M_+$) reduce to a form similar to those describing a two-state exchange process and are given by

$$\frac{d}{dt}\begin{pmatrix} M_+^M \\ M_+^{D1} \end{pmatrix} = \begin{pmatrix} k_{MD}[M] + i\omega_M + R_2 & -k_{DM} \\ -k_{MD}[M] & k_{DM} + i\omega_{D1} + R_2 \end{pmatrix} \begin{pmatrix} M_+^M \\ M_+^{D1} \end{pmatrix}, \qquad (2)$$

where $k_{MD}$ and $k_{DM}$ are the rate constants for formation and dissociation of the dimer, respectively (manuscript in preparation). Hence $p_D$ and $k_{ex}$ extracted from fits of the CPMG dispersion data were interpreted in the context of an asymmetric dimerization process for which $k_{ex} = k_{MD} [M] + k_{DM}$ and $p_D = [D]/([D] + [M]) = k_{MD} [M]/(k_{MD} [M] + k_{DM})$; this allows the determination of the rate ($k_{MD}$ and $k_{DM}$) and equilibrium ($K_{MD}$) constants for the reaction.

It is worth re-emphasizing that on the basis of the residue selection criteria as well as the concentration- and temperature-dependent CEST and CPMG measurements, as described above and summarized in *Figure 8* and *Table 2* of the main text, it is possible to separate the four processes from each other in an unequivocal manner (see *Figures 5, 6*, *Figure 5—figure supplement 1*, *Figure 7—figure supplement 1*).

## Choice of dimer models for the association processes

In the analysis of CEST and CPMG relaxation data as a prelude to obtaining structural models of the oligomers formed via processes I, III and IV we have made the simplest assumption that in all cases dimeric structures are generated. This is justified as follows: (i) the chemical shifts of the excited state conformer formed from process I correlate very well with shifts of mature, native $Cu_2Zn_2SOD1^{S-S}$ published previously (*Strange et al., 2003*) (*Figure 3*), providing very strong evidence that process I corresponds to dimerization of $apoSOD1^{2SH}$ monomers. (ii) Residues involved in the generation of the excited state of process III localize to a very specific region of structure (*Figure 9—figure supplement 1*). Formation of a symmetric dimer with each monomer contributing the same interfacial region can account for the shift changes. More complex models of oligomerization need not be invoked, but if they are then additional interfaces would have to be involved. This is not consistent with the localization of chemical shift changes to only a specific region of structure. (iii) Residues involved in the interface of the oligomer generated from process IV localize to two distinct regions and the chemical shift perturbations can be explained by the formation of an asymmetric dimer, as the simplest model (*Figure 9—figure supplement 1*, *Figure 10*). This does not prove that the structure is dimeric and as is shown in *Figure 9—figure supplement 2*, more complex models involving additional subunits can be generated simply by extending the basic dimeric building block using the same interfaces as those involved in formation of the various dimers via processes I, III and IV. It should be noted that dimeric structures formed in all cases are able to satisfy the input restraints and that more complex models are not required to explain the data.

## Building structural models of excited states using HADDOCK (*Dominguez et al., 2003*)

Residues for which chemical shift changes could be measured in fits of CEST and CPMG experiments were used to generate structural models of the excited states associated with processes I, III and IV. Residues belonging to process I with $|\Delta\varpi_{MD}| > 0.3$ ppm, as established from $^{15}N$, $^{13}C'$ or $^{13}C\alpha$ CEST measurements, were chosen. All residues identified on the basis of the concentration- and temperature-dependence of their $^{15}N$ or methyl-$^{13}C$ $R_{ex}$ values (calculated as described above) were used as 'restraints' in the calculation of excited states for processes III and IV. A list of residues is supplied in *Table 3* (see 'Separating probes of processes I–IV' above for selection criteria of residues to each of the different processes). The solvent accessible subsets of residues from the lists generated as described above were determined using the software package NACCESS (*Hubbard and Thornton, 1993*) using 40% exposure of either backbone or sidechain atoms as the selection criterion; residues in this subset were chosen as 'active' in the HADDOCK docking protocol. One monomer from the $Cu_2Zn_2SOD1^{S-S}$ crystal structure (*Strange et al., 2003*) was chosen as the starting structure in the generation of excited state dimers, since the folds of $apoSOD1^{2SH}$ and each of the monomers of the mature dimer are similar; both have intact β-barrels (*Figure 1*) and differ predominantly in the arrangement and dynamics of the loops. The Zn-loop residues (49-80) and the electrostatic loop (126–143) were fixed to the coordinates of the crystal structure in initial trial runs, but subsequently defined as completely flexible segments to generate dimers for processes III and IV. For process I, only the electrostatic loop was allowed to be fully flexible because chemical shifts indicate that the Zn-loop becomes ordered upon forming the excited state dimer (*Figure 2*). In all HADDOCK runs, histidine protonation states were set using the software WhatIf (*Vriend, 1990*). Docking was done in three stages starting with an initial rigid body step, followed by semi-flexible simulated annealing and a final refinement in explicit water. 1000 structures were generated in the initial rigid-body docking step and the 200 lowest energy structures were chosen for subsequent annealing and refinement.

## Acknowledgements

This work was supported by grants from the CIHR (EMM and LEK). LEK holds a Canada Research Chair in Biochemistry. Grid search computations were performed on the gpc supercomputer at the SciNet

HPC Consortium. SciNet (*Loken et al., 2010*) is funded by: the Canada Foundation for Innovation under the auspices of Compute Canada; the Government of Ontario; Ontario Research Fund—Research Excellence; and the University of Toronto.

## Additional information

### Funding

| Funder | Author |
| --- | --- |
| Canadian Institutes of Health Research (Instituts de recherche en santé du Canada) | Elizabeth M Meiering, Lewis E Kay |
| Canada Foundation for Innovation (Fondation canadienne pour l'innovation) | Lewis E Kay |
| Ontario Ministry of Economic Development and Innovation (Ministre Du Développement Économique Et De L'innovation) | Lewis E Kay |
| University of Toronto (UofT) | Lewis E Kay |

The funders had no role in study design, data collection and interpretation, or the decision to submit the work for publication.

### Author contributions

AS, JAOR, HRB, CMD, EMM, LEK, Conception and design, Acquisition of data, Analysis and interpretation of data, Drafting or revising the article; GB, Analysis and interpretation of data, Drafting or revising the article

## Additional files

### Supplementary files

• Supplementary file 1. Coordinates for the conformationally excited state corresponding to the native dimer (process I); ten structural models with the lowest HADDOCK scores.

• Supplementary file 2. Coordinates for the conformationally excited state corresponding to the symmetric non-native dimer 1 (process III); ten structural models with the lowest HADDOCK scores.

• Supplementary file 3. Coordinates for the conformationally excited state corresponding to the asymmetric non-native dimer 2 (process IV); ten structural models with the lowest HADDOCK scores.

### Major datasets

The following dataset was generated:

| Author(s) | Year | Dataset title | Dataset ID and/or URL | Database, license, and accessibility information |
| --- | --- | --- | --- | --- |
| Sekhar A, Rumfeldt JAO, Broom HR, Doyle CM, Bouvignies G, Meiering EM, Kay LE | 2015 | NMR chemical shifts of pWT apoSOD12SH | http://www.bmrb.wisc. edu/data_library/ summary/index.php? bmrbId=26570 | Publicly available at the Biological Magnetic Resonance Data Bank (26570). |

The following previously published datasets were used:

| Author(s) | Year | Dataset title | Dataset ID and/or URL | Database, license, and accessibility information |
| --- | --- | --- | --- | --- |
| Banci L, Bertini I, Cramaro F, Del Conte R, Viezzoli MS | 2003 | Solution structure of apo Cu, Zn Superoxide Dismutase: role of metal ions in protein folding | http://www.rcsb.org/pdb/ explore/explore.do? structureId=1RK7 | Publicly available at RCSB Protein Data Bank (1RK7). |

| Author(s) | Year | Dataset title | Dataset ID and/or URL | Database, license, and accessibility information |
|---|---|---|---|---|
| Strange RW, Antonyuk S, Hough MA, Doucette P, Rodriguez J, Hart PJ, Hayward LJ, Valentine JS, Hasnain SS | 2003 | The structure of holo type human Cu, Zn superoxide dismutase | http://www.rcsb.org/pdb/explore/explore.do?structureId=1HL5 | Publicly available at RCSB Protein Data Bank (1HL5). |

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
