## [Decision Letter]

Thank you for sending your work entitled “Thermal Fluctuations of Immature SOD1 Lead to Separate Folding and Misfolding Pathways” for consideration at *eLife*. Your article has been favorably evaluated by John Kuriyan (Senior editor) and three reviewers, one of whom, Volker Dötsch, is a member of our Board of Reviewing Editors.

The Reviewing editor and the other reviewers discussed their comments before we reached this decision, and the Reviewing editor has assembled the following comments to help you prepare a revised submission.

This is a beautiful work that provides unprecedented atomic insight into the conformational landscape of the immature SOD1 protein (apoSOD1^2SH^). The system is of exceptional biomedical importance given its involvement in ALS. The authors used an integrated NMR approach spearheaded by CEST and CPMG techniques that provide structural, kinetic and thermodynamic information on alternate, lowly populated conformational states. These very detailed experiments revealed, for the first time, that apoSOD1^2SH^ undergoes at least 4 distinct conformational exchange processes, three of which involve protein dimerization and the other one an intramolecular helix folding transition. Very interestingly, two of the dimerization processes involve the formation of non-native dimers. Untangling such a complex conformational equilibrium (summarized in Figure 8) is extremely challenging. In addition to obtaining kinetics and populations of the excited states, they also determined the chemical shifts of these states and using docking programs they were able to obtain structural models of the various conformational states.

However, every description of a complex system requires some simplification and regarding the justification of these simplifications some questions remain:

1) For process I, the excited state chemical shifts, derived from the CEST spectra, resemble those of the native dimer. While it is clear that the CEST data do report on native dimer formation, the fact that they can be fitted neither to two- or three-state exchange models underscores the complexity of the system and the difficulty in determining accurate kinetic parameters. Some discussion about additional processes that may contribute to the exchange behavior is needed.

2) For process II, the authors invoke a mechanism involving helix formation within the electrostatic loop. They argue that this process is independent of all other processes for the molecule. However, they are unable to model this process as 2-state and again resort to 3-state fitting. In this case the authors appear to fit only linear three-state models. In the SI they show two such schemes. It is not clear which one of these schemes is correct, however only the second scheme (off-pathway) is mentioned in the text and no explanation of this behavior is offered. Might it be due to isomerization of proline 62, with the cis and trans states forming helix at different rates? A number of residues around Pro62 show splitting in the HSQC spectrum. The authors observe that the mutation G85R modifies both process II and IV; does this indicate that these two processes are not independent? Is it possible that the rate of formation of the asymmetric dimer via process IV depends on the presence or absence of the electrostatic loop helix?

3) The kinetic scheme shown in Figure 8 is oversimplified and none of the processes are truly independent; the monomer exchanges at comparable rates with all of the dimer states and with the helical state and most residues would therefore be expected to exhibit very complex multi-state exchange behavior. The kinetic scheme in Figure 8 seems to suggest that native helix formation in the monomer (process II) precludes the formation of any dimer, including the native dimer. Can the helical state of the monomer dimerize? If not, then why is helix formation not also observed in process I? The processes that form the three dimer species are not independent but are mutually exclusive in that the formation of one dimer means that a different dimer does not form. Further, given that all the dimers are simply rotations of one of the subunits there is no reason to believe that they cannot directly exchange with one another. In this case a much more complex cubic kinetic scheme would be more appropriate than Figure 8.

4) The deconvolution of the dispersion and CEST data to assign residues to the different exchange processes and identify contact sites is not well described and is not entirely convincing. Given the numerous processes involved (Figure 8), it would appear that much of the protein would show multi-state exchange profiles. Identification of chemical shift changes that are uniquely associated with individual processes is difficult and assignment of contact restraints is questionable. Is it not possible that some of the shift changes reflect conformational changes rather than direct contacts within the dimer states? How can the authors justify removal of the restraint for L67, simply because it is violated in structure calculations for non-native dimer 1? This appears to be the only active restraint for process III that differentiates the non-native dimer 1 structure from that of the native dimer formed by process I ([Supplementary-material SD1-data]). A better explanation is required as to how the structure of non-native dimer 1 was determined when the only active restraint was persistently violated. Many of the residues that apparently report on process III are not used as restraints in the docking calculation (compare Figure 5 and Figure 8–figure supplement 1). Why is this? This comment also applies to process IV (compare Figure 6 and Figure 8–figure supplement 1). The residues identified with process IV in Figure 6 appear to be scattered over much of the protein surface and there seems to be more going on than formation of a non-native dimer. The authors need to do a better job of describing, in the main text, the constraints used and how they were derived.

5) Figure 9 is misleading, giving the impression that a large number of residues marked with spheres were used to determine the dimer structures. A more informative figure would show all of the residues that are associated with each process and color code those residues that were used as active restraints in the HADDOCK calculations. This would provide the reader with an immediate visual assessment of the extent and quality of data used in the structure calculations; at the moment, this information is available only in the supplemental table.

6) A final concern is that the paper seems to gloss over the asymmetry of the non-native SOD1 dimer 2. It seems from Figure 9 that this is an asymmetric dimer. For asymmetric dimers the resonances of residues located in the interface are expected to be split, which should have implications for fitting of CEST and CPMG data.

---

## [Author Response]

*1) For process I, the excited state chemical shifts, derived from the CEST spectra, resemble those of the native dimer. While it is clear that the CEST data do report on native dimer formation, the fact that they can be fitted neither to two- or three-state exchange models underscores the complexity of the system and the difficulty in determining accurate kinetic parameters. Some discussion about additional processes that may contribute to the exchange behavior is needed*.

Part of the reason that the native dimer process in pWT is multi-state is because of interference from fluctuating magnetic fields arising in other processes. This is seen from the fact that for G85R apoSOD1^2SH^, where process II is eliminated and process IV is reduced, the CPMG profiles for process I can be nicely fit to two-state. We have added discussion about the exchange behaviour of all processes in a new section in Results, titled ‘Complications arising from multiple exchange processes’. With reference to process I, we add:

“For example, CEST profiles of G51, N53, T54 and G61 clearly establish that these residues report on native dimerization, with shift differences in excess of ∼4 ppm between ground and excited state conformers. […]The inability to fit the dispersion data for these residues in the pWT protein is, thus, at least in part the result of fluctuating local magnetic fields that arise from a number of simultaneous proximal processes.”

*2) For process II, the authors invoke a mechanism involving helix formation within the electrostatic loop. They argue that this process is independent of all other processes for the molecule. However, they are unable to model this process as 2-state and again resort to 3-state fitting. In this case the authors appear to fit only linear three-state models*.

Our data is not good enough to reliably resolve the triangular model from linear ones and we have a statement to that effect in Materials and methods under the section titled “Three-state fitting of CEST and CPMG data”:

“The CEST and CPMG data were not of sufficiently high quality to distinguish the triangular model with all three states connected from the linear schemes described above, so the simplest linear 3-state schemes were selected.”

*In the SI they show two such schemes. It is not clear which one of these schemes is correct, however only the second scheme (off-pathway) is mentioned in the text and no explanation of this behavior is offered*.

We cannot distinguish between the on-pathway and the off-pathway models with statistical significance, as both models fit the available data equally well. The numbers that we cite for populations and lifetimes in the Results section and Figure 9, as well as now in Table 1, are the mean and standard deviation of the parameter values obtained from the two models. In other words, the reported parameter values are averages over the models. Despite the difference between the two models (on-pathway vs off-pathway), the values for the excited state lifetime (12.4 ms vs 13.6 ms) and populations (2.0 % vs 2.2%) obtained as detailed in the legend for Figure 3—figure supplement 2 are very similar to one another. Indeed, these are the only two parameters we would expect to reliably extract from a three-state fit of CEST profiles where the third state is not explicitly observed as a minor CEST dip but is necessary for obtaining a robust fit.

We have clarified this point in the figure legend for Figure 3—figure supplement 2.

*Might it be due to isomerization of proline 62, with the cis and trans states forming helix at different rates? A number of residues around Pro62 show splitting in the HSQC spectrum. The authors observe that the mutation G85R modifies both process II and IV; does this indicate that these two processes are not independent*?

Proline isomerization is typically too slow to be detected by CEST (k_ex_ > 20 s^-1^) or CPMG (k_ex_ > 100 s^-1^) methods. If indeed Pro isomerization was an obligate on-pathway step for the formation of the helical folded state, CEST dips would not be observed for this process because of the slow rate of exchange between cis and trans Pro. Consequently, we think it is highly unlikely that the three-state behaviour is due to Pro isomerization.

In the A4V mutant, where all three dimerization processes (I, III and IV) have been eliminated, helix formation (process II) can be adequately modeled as a two-state process, suggesting that the observed three-state behaviour in pWT for helix folding is a result of interference between process II and one or more of the dimerization events, which share some common regions.

The G85R mutant affects both process II and process IV, primarily because the mutation is located in a region of the protein capable of affecting both processes. G85, N86 and V87 are residues which show dispersions belonging to process IV. So it is not surprising that the G85R mutation affects process IV. Further, in wild type SOD1, G85, F45 and D124 (in the electrostatic loop) are involved in a hydrogen bonding network, and this provides some rationale for the G85R mutation can lead to the elimination of process II that involves residues in the electrostatic loop.

*Is it possible that the rate of formation of the asymmetric dimer via process IV depends on the presence or absence of the electrostatic loop helix*?

While processes II and IV do have residues in common, we do not think they are thermodynamically coupled, because process IV can be eliminated without affecting process II in the A4V mutant. Moreover, a coupling of the two would suggest that process II should be concentration dependent (dimerization process IV is) and this is not observed. We have now added an additional paragraph in the Discussion discussing the kinetic scheme that we propose (of necessity highly simplified). See response to point 3 immediately following.

*3) The kinetic scheme shown in*
Figure 8
*is oversimplified and none of the processes are truly independent; the monomer exchanges at comparable rates with all of the dimer states and with the helical state and most residues would therefore be expected to exhibit very complex multi-state exchange behavior. The kinetic scheme in*
Figure 8
*seems to suggest that native helix formation in the monomer (process II) precludes the formation of any dimer, including the native dimer. Can the helical state of the monomer dimerize? If not, then why is helix formation not also observed in process I? The processes that form the three dimer species are not independent but are mutually exclusive in that the formation of one dimer means that a different dimer does not form. Further, given that all the dimers are simply rotations of one of the subunits there is no reason to believe that they cannot directly exchange with one another. In this case a much more complex cubic kinetic scheme would be more appropriate than*
Figure 8.

We have elaborated on the kinetic scheme and our reasons for presenting the exchange processes as distinct events in the Discussion section in a separate paragraph where we state:

“The combination of CEST and CPMG experiments described above leads to a kinetic scheme (Figure 9), in which native apoSOD1^2SH^ appears as a central hub for exchange processes leading to maturation (processes I, II) or aberrant association (processes III, IV). […] these processes involve populations of conformers that are below the detection limits of CEST and CPMG methods.”

*4) The deconvolution of the dispersion and CEST data to assign residues to the different exchange processes and identify contact sites is not well described and is not entirely convincing. Given the numerous processes involved (*Figure 8*), it would appear that much of the protein would show multi-state exchange profiles. Identification of chemical shift changes that are uniquely associated with individual processes is difficult and assignment of contact restraints is questionable*.

We provided a detailed description of how residues were sorted into different processes in the Materials and methods section. In order to further clarify this point, we have added a new figure, Figure 8, summarizing the essential details involved in identifying the residues belonging to each process. We have also added Table 2, which summarizes how processes were distinguished from one another and complements the details provided in Discussion, Materials and methods and in Figure 7—figure supplement 1 and its associated legend.

Indeed conformational exchange in apoSOD1^2SH^ is complicated by the presence of multiple processes and the fact that probes of the different processes can be proximal. However, we have used a combination of complementary experiments, conditions (concentrations and temperatures) and mutants to tease out the residues belonging to each exchange process and define restraints for structural modelling (as described in Figure 8, Table 2, Results and Discussion, and Materials and methods). The assignment of restraints is further supported by the robust characterization of each non-native dimer interface, as seen from the small number of restraint violations (Figure 9—figure supplement 1) and in controls where essentially identical structures are recovered when one restraint is left out at a time in the HADDOCK structure calculations (legend to Table 3).

*Is it not possible that some of the shift changes reflect conformational changes rather than direct contacts within the dimer states? How can the authors justify removal of the restraint for L67, simply because it is violated in structure calculations for non-native dimer 1*?

This is definitely a possibility and a drawback of using chemical shifts as input restraints in HADDOCK. We believe that L67 is changing in chemical shift during process III not because it is at the interface but because it is either sensing fluctuating fields in the vicinity or is a part of a secondary conformational/allosteric effect resulting from dimer formation. We have clarified this point in the legend of Figure 9—figure supplement 1 where we state:

“It appears that spin probes from L67 are not localized to the non-native dimer interface but rather report on a secondary process that depends on dimerization, such as a change in conformation.”

*This appears to be the only active restraint for process III that differentiates the non-native dimer 1 structure from that of the native dimer formed by process I (*[Supplementary-material SD1-data]*). A better explanation is required as to how the structure of non-native dimer 1 was determined when the only active restraint was persistently violated*.

First, L67 is not the only active restraint. There are 12 other active restraints (2 x 6) (Table 3). The reviewers are correct in stating that L67 is indeed one of the few active residues in process III that is not present in process I (I17 is another one – it is not active in Process I) and was identified using the L67Cδ methyl resonance. There are two reasons why L67 was not considered in process I. Firstly, it does not bury significant surface area at the native dimer interface, which is the criterion for selecting a residue to report on process I. Secondly, the amide resonance of L67 is in an overlapped region of the ^1^H-^15^N HSQC spectrum and reliable CEST profiles of L67 could not be obtained. However, we have now recorded ^15^N CEST profiles of L67 by resolving the spectral overlap in a third ^13^CO dimension (manuscript in preparation). We find that the excited state ^15^N chemical shift of L67 matches well with the value in holo SOD, confirming that L67 also responds to process I. Thus, L67 is a probe of both processes I and III.

We have added a paragraph in Results under the section ‘Structural models of excited states of pWT apoSOD1^2SH^’ to elaborate on why the structure of non-native dimer 1 is different from the native dimer, considering that the restraints from process III are essentially a subset of process I. The explanation turns out to be quite simple.

“Notably, the restraints used to model the excited state formed via this process (Table 3 and Figure 9—figure supplement 1) localize to the native dimer interface on one side of the monomer […], leading to a structure where one of the monomers is rotated 180^o^ compared to the native dimer.”

*Many of the residues that apparently report on process III are not used as restraints in the docking calculation (compare*
Figure 5
*and Figure 8–figure supplement 1). Why is this? This comment also applies to process IV (compare*
Figure 6
*and Figure 8–figure supplement 1)*.

This is because many of the residues responding to the dimerization event are not solvent exposed and cannot be used to define the interface in the form of a HADDOCK restraint, as only solvent accessible residues are used as restraints in HADDOCK. In the new Figure 8, we now explicitly state how residues for each process were selected, and how restraints for HADDOCK were chosen from among the selected residues.

*The residues identified with process IV in*
Figure 6
*appear to be scattered over much of the protein surface and there seems to be more going on than formation of a non-native dimer*.

The residues identified for process IV are indeed scattered all over the protein, and the restraints derived from these residues cannot be satisfied by a symmetric dimer. However the restraints are adequately satisfied in an asymmetric dimer, wherein a fraction of the restraints localized to one part of the molecule contribute to the interface of one of the monomers in the asymmetric dimer, while the remaining fraction forms the interface of the other monomer. We have elaborated on this point in the new version of the paper by adding the following in the ‘Structural models of excited states of pWT apoSOD1^2SH^’ section:

“Unlike both dimeric structures formed via processes I and III, the second non-native dimer (non-native dimer 2, process IV) is asymmetric […] that brings one cluster on one monomer close to the remaining clusters on a second molecule.”

*The authors need to do a better job of describing, in the main text, the constraints used and how they were derived*.

We have now added a new figure and a new table, Figure 8 and Table 2, to the main manuscript to summarize the details of how residues were selected for each process and how restraints were picked from the selected residues. This complements the detailed discussion in the Materials and methods section.

*5)*
Figure 9
*is misleading, giving the impression that a large number of residues marked with spheres were used to determine the dimer structures. A more informative figure would show all of the residues that are associated with each process and color code those residues that were used as active restraints in the HADDOCK calculations. This would provide the reader with an immediate visual assessment of the extent and quality of data used in the structure calculations; at the moment, this information is available only in the supplemental table*.

Figure 9 (now Figure 10) is a key figure intended to highlight the regions of the protein involved in the interface of dimers from processes I, III and IV. We believe this figure is important in demonstrating crucial structural features of the non-native dimers in juxtaposition to the native one, and to bring across the point that native dimerization can effectively chaperone SOD1 from forming aberrant interactions. Consequently, we have opted to retain Figure 9 as is. However, we wish to point out in response to the above comment that we do have figures detailing the residues involved in the non-native processes shown on the structure of SOD1 (panel B of Figures 5 and 6) as well as similar figures showing only the active and passive restraints (Figure 9—figure supplement 1 panel A) relevant for each dimer process, since we also considered this information important. We chose not to show residues selected as parts of processes I and II because they were chosen based on our initial idea of what each process was.

*6) A final concern is that the paper seems to gloss over the asymmetry of the non-native SOD1 dimer 2. It seems from*
Figure 9
*that this is an asymmetric dimer. For asymmetric dimers the resonances of residues located in the interface are expected to be split, which should have implications for fitting of CEST and CPMG data*.

Non-native dimer 2 is indeed asymmetric. In order to clarify the implications of formation of an asymmetric dimer on the Bloch-McConnell equations relevant for fitting CEST and CPMG data, we have added a paragraph in the Materials and methods section ‘Separating probes of processes I-IV’ subsection ‘Process IV: transient non-native dimer II formation’ where we state:

“In general, asymmetric dimerization is described in terms of a three-state exchange process, whereby a particular nucleus can have different chemical shifts in the monomer (ω_M_) and in each of the monomers that comprise the dimer (ω_D1_, ω_D2_). However, for the particular case where the environment of the spin probe changes in only one of the monomers of the dimer, as in the case of process IV, it is reasonable to assume that ω_D2_ = ω_M_. In this case, and if the differences in intrinsic transverse relaxation rates of the monomer and dimer are not significantly different, the three-state Bloch-McConnell equations for transverse magnetization (M_+_) reduce to a form similar to those describing a two-state exchange process and are given by[2]ddt(M+MM+D1)=(kMD[M]+iωM+R2−kDM−kMD[M]kDM+iωD1+R2)(M+MM+D1),

where k_MD_ and k_DM_ are the rate constants for formation and dissociation of the dimer, respectively (manuscript in preparation). Hence p_D_ and k_ex_ extracted from fits of the CPMG dispersion data were interpreted in the context of an asymmetric dimerization process for which k_ex_ = k_MD_ [M] + k_DM_ and p_D_ = [D] / ([D] + [M]) = k_MD_ [M] / (k_MD_ [M] + k_DM_); this allows the determination of the rate (k_MD_ and k_DM_) and equilibrium (K_MD_) constants for the reaction.”